# Characterizing the spatiotemporal variability of groundwater levels of alluvial aquifers in different settings using drought indices

Johannes Christoph Haas[1, 2] and Steffen Birk[1, 2]

[1]Institute of Earth Sciences, NAWI Graz Geocenter, University of Graz, Austria
[2]FWF-DK Climate Change, University of Graz, Austria

*Correspondence to:* Johannes Christoph Haas (johannes.haas@uni-graz.at)

**Abstract.** To improve the understanding of how aquifers in different alluvial settings respond to extreme events in a changing environment, we analyze standardized time series of groundwater levels (Standardized Groundwater level Index - SGI), precipitation (Standardized Precipitation Index - SPI), and river stages of three subregions within the catchment of the river Mur (Austria). Using correlation matrices, differences and similarities between the subregions, ranging from the Alpine upstream part of the catchment to its shallow foreland basin, are identified and visualized.

Generally, river stages exhibit the highest correlations with groundwater levels, frequently affecting not only the wells closest to the river, but also more distant parts of the alluvial aquifer. As a result, human impacts on the river are transferred to the aquifer, thus affecting the behavior of groundwater levels. Hence, to avoid misinterpretation of groundwater levels in this type of setting, it is important to account for the river and human impacts on it.

While the river is a controlling factor in all of the subregions, an influence of precipitation is evident too. Except for deep wells found in an upstream Alpine basin, groundwater levels show the highest correlation with a precipitation accumulation period of six months (SPI6). The correlation in the foreland is generally higher than that in the Alpine subregions, thus corresponding to a trend from deeper wells in the Alpine parts of the catchment towards more shallow wells in the foreland.

Extreme events are found to affect the aquifer in different ways. As shown with the well known European 2003 drought and the local 2009 floods, correlations are reduced under flood conditions, but increased under drought. Thus, precipitation, groundwater levels and river stages tend to exhibit uniform behavior under drought conditions, whereas they may show irregular behavior during flood. Similarly, correlations are found to be weaker in years with little snow as compared with those with much snow. This is in agreement with typical aquifer response times over one month, suggesting that short events such as floods will not affect much of the aquifer, whereas a long term event such as a drought or snow rich winter will.

Splitting the time series into periods of 12 years reveals a tendency towards higher correlations in the most recent time period from 1999 to 2010. This time period also shows the highest number of events with SPI values below -2. The SGI values behave in a similar way only in the foreland aquifer, whereas the investigated Alpine aquifers exhibit a contrasting behavior with the highest number of low SGI events in the time before 1986. This is a result of overlying trends and suggests that the groundwater levels within these subregions are more strongly influenced by direct human impacts, e.g. on the river, than by changes in precipitation. Thus, direct human impacts must not be ignored when assessing climate change impacts on alluvial aquifers situated in populated valleys.

# 1   Introduction

Climate change is expected to alter the hydrological cycle and thus the amount and timing of groundwater recharge, storage, and discharge. The future is likely characterized by more extreme hydrological events such as droughts and floods (Seneviratne et al., 2006). Predicting the impact of future climate change on groundwater resources therefore requires a sound understanding of the propagation of extreme events from the atmosphere to the groundwater.

One approach to understanding the variability of groundwater levels is the analysis of the aquifer responses to extreme events in the past (Eltahir and Yeh, 1999; Weider and Boutt, 2010). However, fluctuations of groundwater levels may not only be driven by hydrologic events. In particular, changes in land use or water management are known to be additional important factors (Stoll et al., 2011). Evaluating long-term trends or short-term fluctuations in groundwater level data, therefore, requires careful consideration of the factors potentially controlling the observed changes.

To be able to compare hydrologic extremes between different sites and different types of data various indices have been employed. For instance, the Standardized Precipitation Index SPI (McKee et al., 1993) has been used to identify and analyze the occurrence of extreme events in precipitation. Only recently a corresponding Standardized Groundwater level Index SGI (Bloomfield and Marchant, 2013) has been proposed. SGI values computed for observation wells in the UK (Bloomfield and Marchant, 2013) as well as in Germany and the Netherlands (Kumar et al., 2016) show significant correlation with SPI values. However, the maximum correlation and SPI accumulation period are found to differ between the sites. Thus, as noted by the authors of both studies, groundwater levels and SGI values are influenced by the local hydrogeological conditions.

This work aims to identify factors controlling SGI values of alluvial aquifers within a mountainous region and its foreland (Mur valley, Austria). In this type of setting, groundwater levels measured in the vicinity of rivers are expected to show correlations with the river stage. Going beyond earlier work, therefore, variations of standardized river stages are considered in addition to SPI and SGI. To decipher influences of the local as well as the regional hydrogeological setting correlations between the standardized hydrological time series within three subregions are evaluated and compared with each other. In addition, distinct drought and flood periods as well as a snow rich and snow poor year are analyzed separately, as groundwater levels are known to respond in different ways to floods and droughts (Eltahir and Yeh, 1999). Similarly, one may expect that groundwater levels respond in different ways to abundant and deficient snowfall. Finally, the time series are split-up in several multi-year periods to identify potential long-term changes in the correlations between groundwater levels, precipitation, and river stages.

For this purpose, a novel approach employing correlation matrices is proposed. We visualize these subregions, showing how they differ from each other, how the different bodies of water are related to one another, how they respond to extreme events and how the dynamics in the systems changes over time. We use this approach to select single wells and discuss the limitations of this approach.

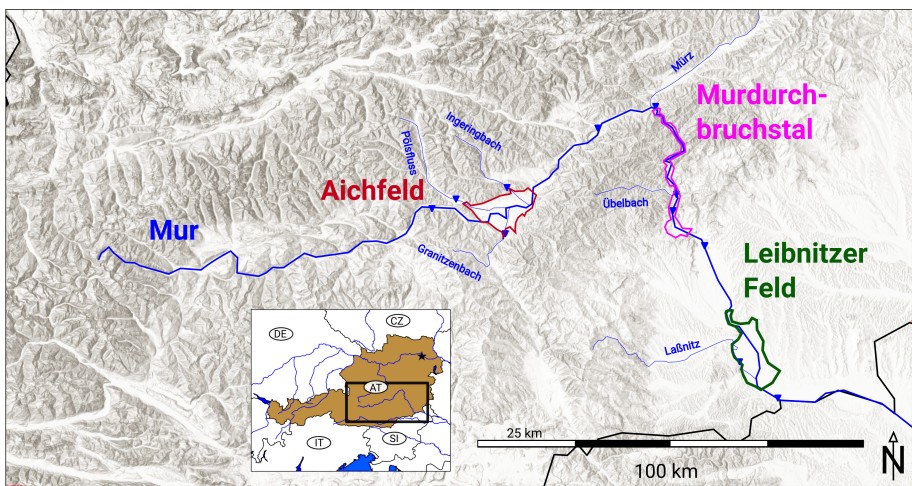

**Figure 1.** Map of the Austrian Mur catchment and its position within Austria, with the subregions studied in detail. See Appendix B for detailed maps of the subregions.

## 2 Method

### 2.1 Study areas

The catchment of the river Mur (Austria) ranges over 300 km from its Alpine source area at 2000 m asl to the Austrian-Slovenian border at 200 m asl (Figure 1). Three distinctive subregions, deemed to differ in their hydrological and hydroge-
5 ological situation, namely the Alpine Aichfeld region, a large and deep basin, the Murdurchbruchstal, a very narrow valley, with small and shallow aquifer bodies and the Leibnitzer Feld, a shallow, mostly river distant lowland aquifer in the Mediterranean/Pannonian climate border region, have been selected for closer investigation.

For these three subregions monthly groundwater levels as well as river stages and precipitation are available at a the ehyd.gv.at website (BMLFUW, 2016). According to the local government agency (personal communication), the data set
started at private house wells, which used to be a common form of water supply in rural Austria. Thus, most of the monitoring wells are assumed to be influenced by human activities. The ehyd.gv.at website provides access to the data of (as of 2015) 950 precipitation measurement stations, 800 surface water gauging stations and 3040 groundwater wells as well as some further hydrological measurements for the whole of Austria (BMLFUW, 2015). The underlying data is managed and quality controlled by the Austrian ministry for agriculture, forestry, environment and water management (BMLFUW). According to
Müller (2006) systematic observation of groundwater began in 1955 with a comparably small number of measuring wells, with the strongest increase in well numbers from 1981 to 1991. In the 1980s, the observations got digitalised and in 1997, digital dataloggers and quality control were introduced into the system. Most of the measurements are taken weekly by hand, but wells are increasingly equipped with dataloggers. In order to assure the quality of the data, various quality controls are conducted before adding it to the database (Godina, 2000; Müller, 2006); (BMLFUW, 2016).

Detailed maps of the following subregions are available in Appendix B. Locations mentioned in the description are marked in said maps. The data sets mentioned are listed in detail in the supplementary material.

### 2.1.1 Aichfeld

The Aichfeld (also called Judenburg-Knittelfelder-Becken) is a large basin in the upper Mur valley. It covers an average elevation of about 650 m asl and an area of around 70 km$^2$. The basin itself is of Tertiary age and contains economic amounts of coal in depths of up to 1000 m bgl (Worsch, 1963). Those have been exploited starting in the 17th century and with industrial underground mining from approx. 1860 to 1978, in the town of Fohnsdorf, in the north-west of the basin (Scheucher, 2004). Above its deep basin fill of Tertiary shales, marls and sandstones, it is filled with around 70 m of fluvio-glacial sediment - mostly gravels and sands, with significant clay layers only in some areas - in a terraced structure and surrounded by mountainous area of elevations between 1500 m asl and 2400 m asl. Arbeiter et al. (1980) lists hydraulic conductivities for 9 locations in the subregion obtained from pumping tests conducted between 1975 and 1977. The conductivities range from $6.7 \times 10^{-4}$ m/s to $1.1 \times 10^{-2}$ m/s, with their mean at $6.3 \times 10^{-3}$ m/s. The surveyed sand and gravel aquifer has an average thickness of 16.4 m and is covered by loamy and fine sands varying between 0.6 and 2 m thickness. The average saturated thickness is 14 m, suggesting generally unconfined conditions.

Climatically, due to its basin structure, the region is prone to inversion climates with strong nightly cooling. For the climate station Zeltweg - in the center of the basin - ZAMG (2016) gives an average yearly temperature of 6.6 °C, an average yearly precipitation of 800 mm and an average 75 cm of snowfall (1971 - 2000).

The towns in the Aichfeld form an Alpine agglomeration with about 50000 inhabitants in the basin and about 80000 taking the surrounding catchment into account. Given this population and the associated settlement history and industry density, the area has a considerable infrastructure of groundwater wells, starting with the Knittelfeld drinking water supply from 1899 on (Gemeinde Knittelfeld, 2016), and considerable drainage activities during the days of active coal mining.

The data set for the Aichfeld consists of 20 groundwater monitoring wells (see supplementary material) covering the time span from 1975 to 2010. The surface elevations range from 693 to 619 m asl and the average depth of the wells below ground level is 13.5 m with a high standard deviation of 8.5 m, which can be explained by the existence of two aquifers, a shallow one and a deep one (see Section 3.1.1). A visual survey of aerial photography for the area shows that only 1 of the 20 wells is not in the close vicinity of farm, residential or industrial buildings, so direct human influence on most wells is likely. The river Mur in the Aichfeld region is only used by three small-scale run-of-the-river hydro power plants in its upstream part. So only 3 wells are situated in the vicinity of a stretch of the river that is deemed impounded. Consequently, the average distance from a well to an upstream power plant is 5.6 km, whereas the downstream distance - mostly to a power plant outside of the subregion - is 26 km.

Out of this data set of 20 wells, 3 wells were selected for closer investigation (see Table 1 and Figure 2).

### 2.1.2 Murdurchbruchstal

The Murdurchbruchstal is a narrow valley, where the Mur leaves the Mur-Mürz Furche and cuts through a mountain range, thus forming a mostly very narrow and steep valley until it reaches the lowlands south of Graz. This subregion covers an area of around 41 km$^2$ and an elevation from approx. 480 m asl at the town of Bruck an der Mur at the beginning of the valley to approx. 368 m asl at the outskirts of the city of Graz at the end of the valley.

From the town of Bruck an der Mur at the beginning, the valley is incised into metamorphic gneisses, amphibolites and shists of the Austroalpine crystalline basement. At the town of Mixnitz, roughly in the upper third of the subregion, this changes to the shales and mostly limestones of the Paleozoic of Graz, that is forming the Central Styrian Karst and the Graz Highlands (Wagner et al., 2011). This change in geology is also reflected in the structure of the aquifer, where considerable aquifer bodies are only found downstream of Mixnitz (Anderle, 1969).

The valley itself is filled with various, mostly unconsolidated sediments. According to Zetinigg et al. (1966), these are mostly postglacial riverine gravels, some old glacial terraces at the margins of the valley, the alluvial fans of tributaries and weathered slope rock, all covered in part by clays. For the 2 km$^2$ location of Friesach in the lower part of the subregion, Zetinigg et al. (1966) lists thicknesses of 8 to 27 m for the central valley fill gravels. For the whole valley Anderle (1969) also states that the aquifer thickness is "very variable" with a saturated thickness between 15 and 20 m. The water level is close to the surface (0 - 4 m depth to water table) and covered by 1 - 1.5 m loamy, fine sands in the areas close to the river Mur, whereas the cover can extend to a thickness of 4 to 15 m of gravels and sands in the terraces and fans at the margins of the valley, suggesting mostly unconfined conditions. The only hydraulic conductivity estimate available for the area is a value of $1 \times 10^{-3}$ m/s from a pumping test near the town of Judendorf-Straßengel in the lower part of the subregion (Zetinigg, 1982).

No climate data is available in the Murdurchbruchstal itself, but ZAMG (2016) provides information for the station in Bruck an der Mur at the beginning of the valley, where an eastern Alpine valley climate with low winds prevails. The average yearly temperature is 8.1 °C, the average yearly precipitation is 795 mm, with an average of 73 cm of snowfall (1971 - 2000).

The settlements in the area are mostly small, though with considerable industries (quarries, paper production) in some locations and a chain of 8 run-of-the-river hydro power plants over a valley length of approx. 30 km, turning large parts of the river into storage area for said power plants. Further, there is a large water plant for the city of Graz in the vicinity of the town of Friesach, where extraction of drinking water is conducted since 1977 as well as infiltration of river water from 1980-1982 on and there are communal water plants at the towns of Gratwein, Judendorf-Straßengel and Gratkorn (Benischke et al., 2002), (ÖVGW, 2016).

The data set for the Murdurchbruchstal consists of 24 groundwater monitoring wells (see supplementary material) covering the time span from 1980 to 2010. The surface elevations range from 413 to 374 m asl and the average depth of the wells below ground level is 10.7 m with a standard deviation of 4.3 m. Due to their vicinity to buildings, 16 of the 24 well are considered likely to be directly human influenced. With the 8 large hydro power plants in the subregion, 4 wells are situated in the vicinity of a stretch of river that is impounded, with an additional 10 wells where an influence is considered likely. The average distance from a well to an upstream power plant is 2.4 km and the average distance to a downstream one is 3.2 km.

Out of this data set of 24 wells, 3 wells were picked for closer investigation (see Table 1 and Figure 2).

### 2.1.3  Leibnitzer Feld

The Leibnitzer Feld is a large and topographically relatively flat lowland basin of the river Mur, named after its central town. Important rivers besides the Mur are the Laßnitz and the Sulm in the western part of the basin. Besides the town of Leibnitz, the area is mostly used for agriculture. This subregion covers an area of around 100 km$^2$ and an elevation from approx. 302 m asl at the town of Mellach at the northern tip of the subregion and approx. 258 m asl at the town of Ehrenhausen at the southern tip of the subregion.

The region is underlain by the Neogene Styrian Basin which consists of various layers of sea, lake and river sediments, which are in turn underlain by the continuation of the Paleozoic of Graz. Apart from the Leitha limestones at the town of Wildon at the northern border of the region, all of the Tertiary sediments are very soft, so they have been mostly eroded and replaced with a series of quaternary gravels, sands and clays in a terraced form (Fabiani, 1971). The mentioned limestones at Wildon are narrowing the aquifer and are thus a natural barrier against inflow from upstream, whereas the southern border is well connected to its downstream regions.

The thicknesses of the groundwater bearing gravels in the vicinity of the river Mur is between 4 and 6 m in the north-east of the region and 3 to 5 m in the south-east with coverages of fluvial gravels, sands and clays of only 0 to 3 m, whereas the higher terraces can have aquifer thicknesses of 3 to 6 m with 3 to 10 m of coverage (Fabiani, 1971). In most areas of the subregion, the saturated thickness of the unconfined aquifer is less than 4 m (Fank et al., 1993). Fank et al. (1993) compiled 20 hydraulic conductivity estimates for various locations in the subregion obtained from various reports and pumping tests conducted from 1967 to 1991. The conductivities range from $2.0 \times 10^{-3}$ m/s to $1.6 \times 10^{-2}$ m/s, with their mean at $4.89 \times 10^{-3}$ m/s. Fank et al. (1993) conclude that the differences between the conductivies are "rather small", however there are some areas with highly variable values due to the inhomogenous sedimentation history of the river Mur, e.g. oxbows filled with fine sands or coarse gravels.

According to ZAMG (2016), the town of Leibnitz has an average yearly temperature of 8.8 °C, an average yearly precipitation of 908 mm and 49 cm of snowfall (1971 - 2000).

The data set for the Leibnitzer Feld includes 31 groundwater monitoring wells (see supplementary material) covering a time span from 1975 to 2010. The surface elevations range from 298 to 259 m asl and the average depth of the wells below ground level is 6.4 m with a standard deviation of 2.9 m. Due to their vicinity to buildings, there are only 3 wells where a direct human influence is considered unlikely.

Since the Mur in the Leibnitzer Feld region is also heavily used for power production with 5 run-of-the-river power plants, 9 wells are located in areas where the Mur is clearly impounded, with another 11 wells where this is considered likely, and 8 wells where it is not clearly visible, leaving only 3 wells situated in parts of the area where the river is not impounded. Due to the large extent of the region and the size of the hydro power plants, the average distance from a well to an upstream power plant is 3.2 km and the distance to a downstream power plant is 3.2 km.

Out of this data set of 31 wells, 2 wells were picked for closer investigation (see Table 1 and Figure 2).

**Table 1.** Wells selected for closer investigation or specifically mentioned in the text. The "HZB" (from Hydrographisches Zentralbüro) refers to their identifier at the ehyd.gv.at website. The "Identifier" is a short code used in this paper to identify the wells in the various plots. "Influence" lists factors that might affect the behavior of the groundwater shown in the well.

| Subregion | HZB | Location | Identifier | Influence |
|---|---|---|---|---|
| Aichfeld | 314807 | Aichdorf | AAn | Well located in a deeper aquifer body, only well in the data set that is not located close to human settlements or activities, deepest well in the data set |
| Aichfeld | 315077 | Raßnitz | ARf | Well deviating from the average behavior in the subregion in the 2009 flood year (see Section 3.2 and Figure 3) |
| Aichfeld | 314922 | Apfelberg | AAr | Well closest to the river Mur, very high correlation with SRSI and neighboring wells SGI time series |
| Aichfeld | 211128 | Pölsfluß | APr | Mid sized tributary stream, deemed mostly natural |
| Aichfeld | 211185 | Mur Leoben | AMr | River Mur, gauge downstream of the subregion |
| Murdurchbruchstal | 325506 | Friesach-St.Stefan | MFd | Well deviating from the average behavior in the subregion in the 2003 drought year (see Section 3.2 and Figure 3), located next to the Friesach water plant |
| Murdurchbruchstal | 325142 | Deutsch Feistritz | MDp | Well located close to a power plant, no likely direct human impact besides this |
| Murdurchbruchstal | 325191 | Kleinstübing | MKr | Well without obvious human influence, close to the river |
| Murdurchbruchstal | 328674 | Judendorf-Strassengel | MJc | Well located central in the highly correlated "cluster" in Figure 2 |
| Murdurchbruchstal | 211649 | Übelbach | MUr | Mid sized tributary stream, deemed mostly natural |
| Murdurchbruchstal | 211292 | Mur Bruck | MMr | River Mur, gauge upstream of the subregion |
| Leibnitzer Feld | 311514 | Untergralla | LUr | Well located closest to the river Mur, no directly visible human influence |
| Leibnitzer Feld | 311001 | Joess | LJc | Well highly correlated to most of the other SGI time and the SPI, direct human influence likely, close to river Laßnitz |
| Leibnitzer Feld | 211466 | Mur Spielfeld | LMr | River Mur, gauge downstream of the subregion |
| Leibnitzer Feld | 211441 | Laßnitz | LLr | Mid sized tributary stream, deemed mostly natural |

## 2.2 Drought indices

Monthly time series were obtained for the subregions from ehyd.gv.at (BMLFUW, 2016). Single time series have been used for groundwater monitoring wells and river stage measurements, whereas the precipitation is averaged. Due to the size and topography of the subregions and our approach to work with monthly data, we consider an averaged precipitation over the

subregion as a valid approach. However, some events (such as summer thunderstorms) can be very intense and affect only a very small part of a subregion, so some wells or tributary streams could be affected by such an event that is not accounted for in the average precipitation. Short gaps (only relevant for 1 to 4 wells per subregion) have been padded with the previous water level.

5  Due to the different start and end dates of the single time series, the raw data has been cut to periods offering both the most wells for the subregion in question and the longest possible time period.

To be able to compare both different types of data and different subregions the data was standardized using the Standardized Precipitation Index (SPI, McKee et al. (1993)), the Standardized Groundwater Index (SGI, Bloomfield and Marchant (2013)) and the SGI applied on river stages (SRSI).

### 2.2.1  SPI

For precipitation, the SPI, developed by McKee et al. (1993) is used. This allows for both a standardization of data and the computation of average standardized precipitation, where McKee et al. (1993) suggest averaging For the standardization, the data set gets split-up into time series for each month, which is then fitted to the gamma distribution to relate the respective months to each other instead of months from different seasons.

While there is some criticism of the gamma distribution (see e.g. Guttman (1999) and Blain and Meschiatti (2015)), it is generally a widely used and recommended index (see e.g. Svoboda et al. (2012)).

### 2.2.2  SGI

For the groundwater, the relatively new Standardized Groundwater Index, SGI proposed by Bloomfield and Marchant (2013) has been used. The SGI is based on the SPI, but whereas the SPI uses a fixed transformation of the raw data by fitting it

on a gamma distribution, the SGI uses a non-parametric normal scores transform on the raw data, taking into account the different possible distributions of groundwater time series. Similar to the SPI, the data set gets split-up into time series for each month (e.g. January 1982, January 1983, January 1984, etc. ; February 1982, February 1983, February 1984, etc.) to relate the respective months to each other instead of months from different seasons.

Unlike the SPI, the SGI is not accumulated over specific time periods due to the continuous nature of the underlying ground-

water level (Bloomfield and Marchant, 2013).

### 2.2.3  SRSI

To characterize and monitor hydrological drought, streamflow indices were previously employed (e.g., Vicente-Serrano et al. (2012); Lorenzo-Lacruz et al. (2013); Barker et al. (2016)). As we are interested in the impact of rivers on groundwater level fluctuations, it is straightforward to consider river stages instead of streamflow.

In order to be able to compare river stages with precipitation and groundwater, we used the SGI on river water levels. Due to its self fitting nature, it can also be used with river water levels, which have a probability distribution different from many groundwater times series.

In order to fit with the naming convention of the other indices, we propose to name this index the SRSI - Standardized River Stages Index.

## 2.3 Correlation matrix

For each possible combination of standardized groundwater (SGI), standardized precipitation (SPI) or standardized river stages time series (SRSI) a Pearson correlation coefficient was calculated. In order to facilitate the comparison of standardized groundwater levels, river stages, and precipitation within the individual subregions, the above mentionend Pearson correlation coefficients have been plotted as correlation matrix, showing all the SGI time series, all the SRSI time series and SPI1, 3, 6, 9 and 12 for each subregion, similar to the matrices applied in Stoll et al. (2011) and Loon and Laaha (2015). For a detailed description on how to read correlation matrices, please refer to Appendix A.

According to Vekerdy and Meijerink (1998), highest correlations between daily river stages and groundwater levels in distances similar to those relevant for this paper are mostly found for lag times below 30 days. Likewise, Bloomfield and Marchant (2013) as well as Kumar et al. (2016) found with few exceptions the highest correlation between SGI and SPI associated with a time lag of zero months. Our dataset follows this expectation, with more than 80% of SGI-SPI pairings for the shallow part of the Aichfeld, the Murdurchbruchstal and the Leibnitzer Feld showing the highest Pearson correlation coefficient for a time lag of 0 months. In the cases where the highest correlation coefficient occurs at a time lag other than 0 months, which mainly concerns correlations with the SPI9 and SPI12, most of the differences to the 0 month corrrelation coefficient are neglible (average difference: 0.003 for 6 SPI12-SGI pairings with 1 month lag in the shallow Aichfeld), small (average difference: 0.01 for 19 SPI12-SGI pairings with 1 month lag in the Leibnitzer Feld) or occur at very low correlated time series (6 SPI1-SGI pairings in the shallow Aichfeld with their highest correlation coefficient of < 0.2 occurring at time lags of 36 - 39 months in the Murdurchbruchstal). A similar situation occurs with the SRSI-SGI pairings, where more than 95% have their highest Pearson correlation coefficient at a time lag of 0 months, with the only exceptions being 8 low correlated (r < 0.2) SRSI-SGI pairings with their highest correlation occurring at time lags of 39 - 48 months. Therefore, we consistently apply only Pearson correlation coefficients without a time lag.

## 3 Results

### 3.1 Observations within the subregions

#### 3.1.1 Aichfeld

In the Aichfeld subregion two patterns emerge (Figure 2): A large area in the plot shows SGI time series (standardized groundwater levels measured at different wells) that are highly to very highly correlated with each other and with the SRSI time series

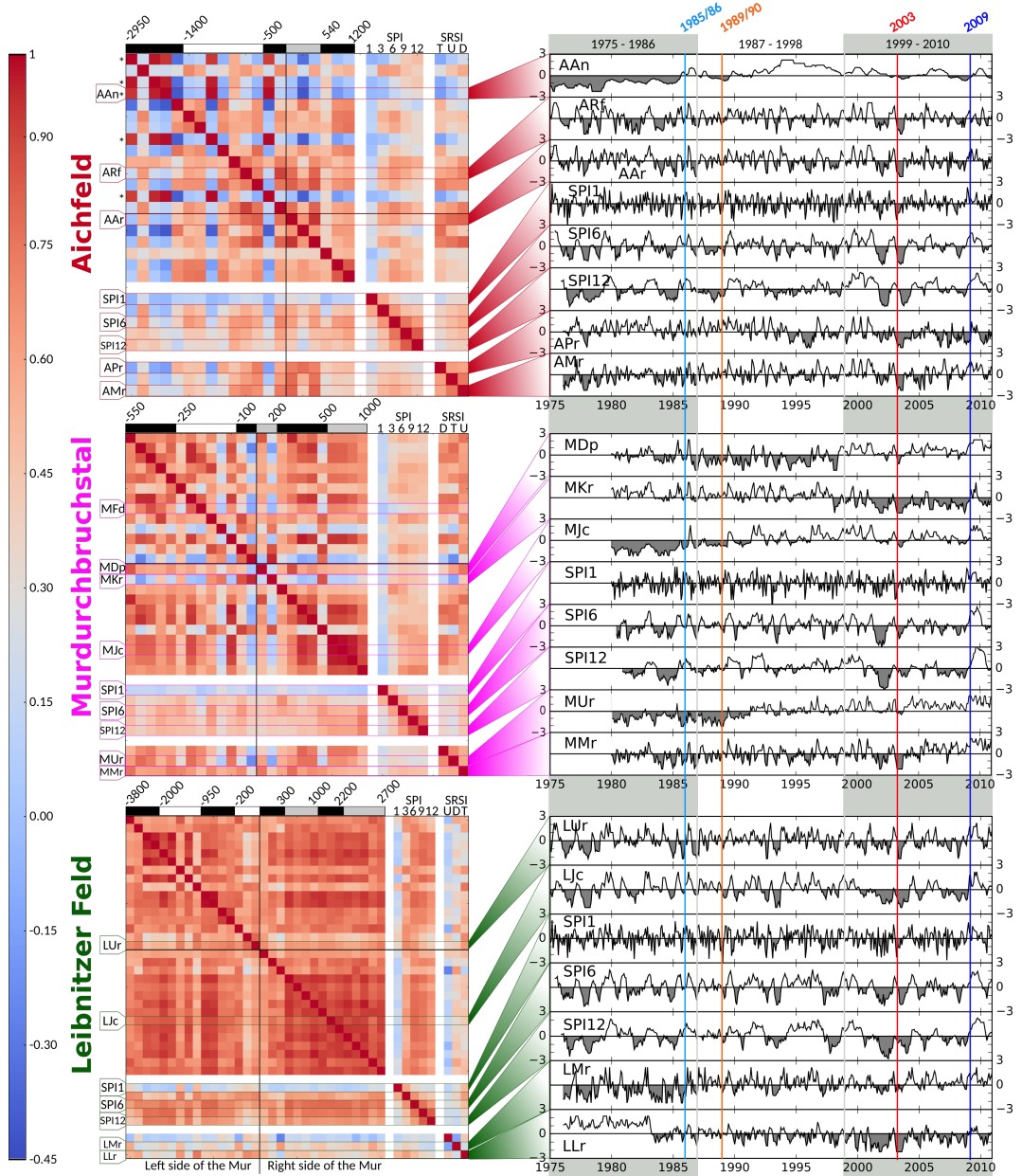

**Figure 2.** Correlation matrices for the three subregions (left side) with selected time series for SGI, SPI and SRSI (right side). The data for each subregion is sorted in three groups, divided by blank columns and rows: 1 - Groundwater, SGI, sorted by distance of the well to the stream, given in meters on the top of the matrices; 2 - Precipitation, SPI1, 3, 6, 9 and 12; 3 - Surface water, SRSI, for different Mur gauges or streams in the subregion (U: Mur upstream, D: Mur downstream, T: Tributary stream). Deep wells in the Aichfeld are marked with an asterisk *. The three letter markers on the left highlight selected wells and river stages discussed in the text. See also Table 1 . Also shown are the time periods used in Section 3.4 and Figure 5, the years 1985/86 and 1989/90 used in Section 3.3 and Figure 4 and the years 2003 and 2009, used in Section 3.2 and Figure 3. For further details on correlation matrices, please refer to Appendix A.

**Table 2.** Average Pearson correlation coefficients of a subregion for SGI time series with each other(SGI with SGI), SGI time series with single SRSI time series (SGI with SRSI 1, ...) and SPI averaging periods (SGI with SPI1, ...) and the average correlation coefficient for all SRSI time series of a subregion with the SPI averaging periods (SRSI with SPI1, ...) with their standard deviations for each subregion.

| Location | SGI with SGI | SRSI 1 | SRSI 2 | SRSI 3 |
|---|---|---|---|---|
| | | Pöls | Mur up | Mur down |
| Aichfeld shallow | 0.59 ±0.15 | 0.55 ±0.14 | 0.50 ±0.16 | 0.52 ±0.19 |
| Aichfeld deep | 0.96 ±0.031 | -0.13 ±0.022 | 0.045 ±0.017 | 0.24 ±0.015 |
| | | Mur down | Übelbach | Mur up |
| Murdurchbruchstal | 0.55 ±0.27 | 0.55 ±0.21 | 0.51 ±0.29 | 0.60 ±0.11 |
| | | Mur up | Mur down | Laßnitz |
| Leibnitzer Feld | 0.73 ±0.14 | 0.16 ±0.15 | 0.38 ±0.11 | 0.44 ±0.14 |

| | SGI with SPI1 | SPI3 | SPI6 | SPI9 | SPI12 |
|---|---|---|---|---|---|
| Aichfeld shallow | 0.15 ±0.085 | 0.47 ±0.13 | 0.57 ±0.11 | 0.47 ±0.12 | 0.38 ±0.12 |
| Aichfeld deep | -0.039 ±0.013 | 0.0049 ±0.023 | 0.19 ±0.048 | 0.32 ±0.077 | 0.38 ±0.081 |
| Murdurchbruchstal | 0.16 ±0.060 | 0.41 ±0.092 | 0.51 ±0.090 | 0.49 ±0.094 | 0.47 ±0.090 |
| Leibnitzer Feld | 0.21 ±0.12 | 0.58 ±0.10 | 0.72 ±0.069 | 0.68 ±0.10 | 0.61 ±0.11 |

| | SRSI with SPI1 | SPI3 | SPI6 | SPI9 | SPI12 |
|---|---|---|---|---|---|
| Aichfeld | 0.27 ±0.083 | 0.45 ±0.077 | 0.48 ±0.088 | 0.42 ±0.086 | 0.34 ±0.074 |
| Murdurchbruchstal | 0.26 ±0.044 | 0.38 ±0.064 | 0.39 ±0.095 | 0.35 ±0.11 | 0.37 ±0.080 |
| Leibnitzer Feld | 0.31 ±0.19 | 0.41 ±0.20 | 0.36 ±0.12 | 0.26 ±0.095 | 0.24 ±0.068 |

(standardized river water levels at measured at different gauging stations) in the subregion. The SGI time series are from the wells situated closest to the river Mur on both riverbanks (represented by well AAr). Most wells outside of the core of this region show a similar behavior, resulting in an average Pearson correlation coefficient of all of these SGI time series with each other of 0.59. These SGI time series show a low correlation with the SPI1 time series and moderate to high correlations with the

5 longer SPI averaging periods, as expected from the previous literature (Bloomfield and Marchant, 2013; Kumar et al., 2016). The average Pearson correlation coefficient of all of these SGI time series with SPI1 is 0.15, which raises to a maximum with SPI6 of 0.57 and decreases to 0.38 with SPI12. The average correlation of the SGI time series with the SRSI time series in the subregion is similar for all river gauging stations, with an average of 0.52 (see "Aichfeld shallow" in Table 2).

The second feature of the region are 5 wells (represented by well AAn in Figure 2) that show a very low to negative

10 correlation of their SGI time series with those of all other wells as well as with all SPI and SRSI time series in the subregion,

but are extremely highly correlated with each other, with an average Pearson correlation coefficient of these SGI time series with each other of 0.96, whereas the remaining wells have an average correlation coefficient with each other of 0.59. This difference in correlations is highly significant ($p < 0.01$, t-test). A look at the underlying data reveals that the wells first discussed reach an end depth significantly deeper (avg. 24.9 m bgl) than that of the other wells in the data set (avg. 9.7 m bgl), so it is reasonable to assume that they show a different, deeper aquifer system. This is also in accordance with Worsch (1963), who mentions that earlier wells of a similar depth for the military airfield at this location encountered a conglomerate layer and Stadlbauer and Lorbeer (2000) who mention a significant groundwater inflow in this area. The wells from the deeper aquifer also show a clear increase of correlation of SGI time series with an increase in the length of the SPI averaging periods, starting with an average correlation of the SGI time series with the SPI1 of -0.04, reaching a maximum correlation of 0.38 with the SPI12, which is significantly lower than the correlations seen in the shallow wells SGI time series. The average correlations of the deeper wells SGI with the SRSI time series range from -0.13 with the local Pöls to 0.24 with the downstream Mur. In the following, the focus is on the shallow groundwater, but in some places we will consider the deep wells for comparison.

The SRSI time series are correlated well with each other, indicating a similar flow regime in the upstream and downstream Mur, as well as in the tributary Pöls, but the correlations with the SPI time series are low to moderate, ranging from an average of 0.27 with SPI1 to 0.48 with SPI6.

For further investigations, one of the wells from the shallow wells with the highly correlated SGI time series and one well from the deeper aquifer have been picked (see also Table 1).

Well AAr, with a SGI time series highly correlated with most other shallow wells SGI time series and closest to the river Mur, shows frequent changes between wet and dry conditions of different lengths and magnitudes just as the highly correlated AMr Mur gauge downstream of the subregion. Generally, this fast changing well shows only moderate correlation with SPI time series, no matter the averaging period. However, large events such as the 2002 and 2003 double drought are clearly visible.

Well AAn, situated in the deeper aquifer system and far away from the river Mur, shows a much slower oscillation of the water levels, overlain by a long-term trend from wet conditions into dry ones and then possibly back into wet. Apart from large events, such as the double wet event in 1985 and 1986 and the double drought in 2002 and 2003, no similarities with the shallow wells, the precipitation or the river gauging stations are obvious.

### 3.1.2 Murdurchbruchstal

In this subregion, the matrix visualization shows a picture noticeably different from the upstream Aichfeld (see Figure 2).

A highly significant ($p < 0.01$) difference is visible when comparing the SGI correlation coefficients of the complete Aichfeld or the deep Aichfeld with the Murdurchbruchstal, which is also reflected in the differences between the average correlation coefficients of 0.39 for the complete Aichfeld, 0.96 for the deep Aichfeld and 0.55 for the Murdurchbruchstal. The shallow Aichfeld shows a groundwater signal similar to the Murdurchbruchstal and thus no significant ($p > 0.05$) change in the correlations. This is also shown by similar average correlation coefficients of 0.55 for the Murdurchbruchstal and 0.59 for the shallow Aichfeld (see also Table 2).

As expected in a narrow valley with aquifers of small spatial extend, there is a high correlation between SGI time series and SRSI time series. A cluster of highly correlated SGI time series (represented by well MJc) is situated at the furthermost distances to the river on its right bank, which are all - except for one well - situated in the town Gratwein-Straßengel and are also highly correlated with the SGI time series of the single well situated in the neighboring town of Gratkorn on the opposite side of the Mur. This cluster and the majority of the SGI time series in the subregion show high to very high correlations with the SRSI time series. The average correlation for the SGI with the SRSI time series is the highest for the upstream Mur gauge with a Pearson correlation coefficient of 0.6 and the lowest for the local tributary Übelbach with 0.5. Correlations with the precipitation are generally lowest with the SPI1 with an average of 0.16 and have the highest correlations with the SPI6 and 9 with average Pearson correlation coefficients of 0.51 and 0.49 respectively.

Surprisingly, some of the wells closest to the Mur on both sides of the river are not very well correlated with each other and are also not among the wells with highest correlations between SGI and SRSI time series. In particular, the matrix view shows three clear outliers (well 4th closest and well closest to the Mur on its left bank and second closest on its right (well MKr)), whose SGI time series are correlated very low or negative with the rest of the SGI time series, but high to very high with each other. The pair of Mur-close wells is situated in the same stretch of the river Mur opposite each other. These wells are also the only wells that have SGI time series negatively correlated with the SRSI time series in the system. For further investigations, one of the wells from the cluster (MJc), the well closest to the river Mur (MDp) and one well from the outliers also very close to the river Mur (MKr) have been picked (see also Table 1).

Well MJc is located centrally in the highly correlated cluster of wells and shows a trend from mostly dry conditions to wetter conditions, which matches the observation of the local tributary Übelbach (MUr). The SPIs for the subregion show no such trends, however the SPI6 and SPI9 show large dry events in the period from 1980 - 1992, as well as the 2003 drought and 2009 flood. Some large events, such as the 2003 drought and 2009 flood are also noticeable in well MJc albeit not too significantly due to the underlying trend from dry conditions to wetter conditions.

Well MKr is located very close to the river Mur, yet it shows no high correlation with it. We observe wet conditions until 1999 and dry conditions thereafter. Large events are also visible in this time series, albeit damped or amplified by the change in conditions around 1999. Well MDp is located very close to well MKr and very close to the river Mur and shows an opposite change from dominant dry conditions until 1999 to wet conditions afterwards. This phenomenon is discussed in detail in Section 4.5.

The river gauges SRSI time series are very highly correlated with each other, but only show some minor correlations with SPI1 (average correlation coefficient 0.26) and SPI3 - 9 (average correlation coefficients 0.35 - 0.39).

### 3.1.3 Leibnitzer Feld

In the Leibnitzer Feld, the situation is different again (see Figure 2). Besides the fact that this region has a much higher amount of groundwater wells, the matrix visualization again shows a very different picture compared with the previous two subregions. These differences in the correlations of the SGI time series in each subregion with each other are highly significant ($p < 0.01$)

and are also reflected in their average correlation coefficients of 0.59 for the shallow Aichfeld, 0.55 for the Murdurchbruchstal and 0.73 for the Leibnitzer Feld.

Apart from a zone of wells with differing SGI time series on both benches of the river (represented by well LUr) and some wells with moderately correlated SGI time series on the left side, high to very high correlations of most SGIs with each other prevail, resulting in an average Pearson correlation coefficient of 0.73. Likewise high correlations of SGI time series with SPI time series can be observed in almost all wells, with the highest correlations found with the 6 and 9 month SPI, with average correlation coefficients of 0.72 and 0.68 respectively. Unlike the other subregions, the correlations of the SGI time series with the SRSI time series are generally low to negative even for the SGI time series of wells very close to the Mur. The lowest average correlation is seen at the upstream Mur with an average of 0.16 and the highest at the local river Laßnitz with 0.44.

It should be noted that part of this can be explained by the fact that the Leibnitzer Feld is also a region where the Mur is heavily used for power production, so the river levels and their fluctuations are not natural. Due to the different times the dams have been built, it is also likely that significant changes in the river regime have occurred during the life time of the data set. In addition, both gauging stations for the Mur used for this subregion are outside of the subregion and outside of the area of influence of the power plants in the subregion, so they likely show a behavior different from that of the river Mur within this subregion.

For further investigations, one of the wells from the highly correlated group and one well close to the river have been picked (see also Table 1).

Well LJc, whose SGI time series is highly correlated to most SGI values in the subregion, shows frequent changes between dry and wet conditions. Compared with the SPI1 or the river gauges LMr and LLr, it shows a smooth signal visually similar to the highly correlated SPI6. Large events such as the two droughts between 1976 and 1979 are also similar to the river Mur (LMr) or river Laßnitz (LLr) in the case of the 2002 and 2003 droughts.

The SGI time series of well LUr, situated right next to the river Mur, shows only moderate correlations with most SGI time series in the subregion. Just as well LJc, it shows frequent changes between dry and wet conditions. The correlation is highest with the SPI3 (not shown in Figure 2), but despite a slightly lower correlation the SPI6 shows a good visual fit with well LUr too. Large events such as the 1976 - 1979 and 2002 - 2003 droughts are visually similar to the river time series LMr and LLr, but apart from that, the river gauging stations show a behavior different from that of the nearby wells.

The mentioned discrepancies in the water levels of the river Mur are also visible in the correlations of the three river gauging stations SRSI time series with each other. Here, unlike in the other regions, generally very low correlations are seen not only when comparing the Mur with the Laßnitz - which is expected due to their different catchments - but also when comparing the two Mur stations, which would be expected to show a similar signal, if they where behaving naturally. Only the local tributary Laßnitz shows a moderate correlation with the 1 to 3 months SPI. For the average correlations with the SRSI time series, the highest value is seen for the SPI3 with a Pearson correlation coefficient of 0.41.

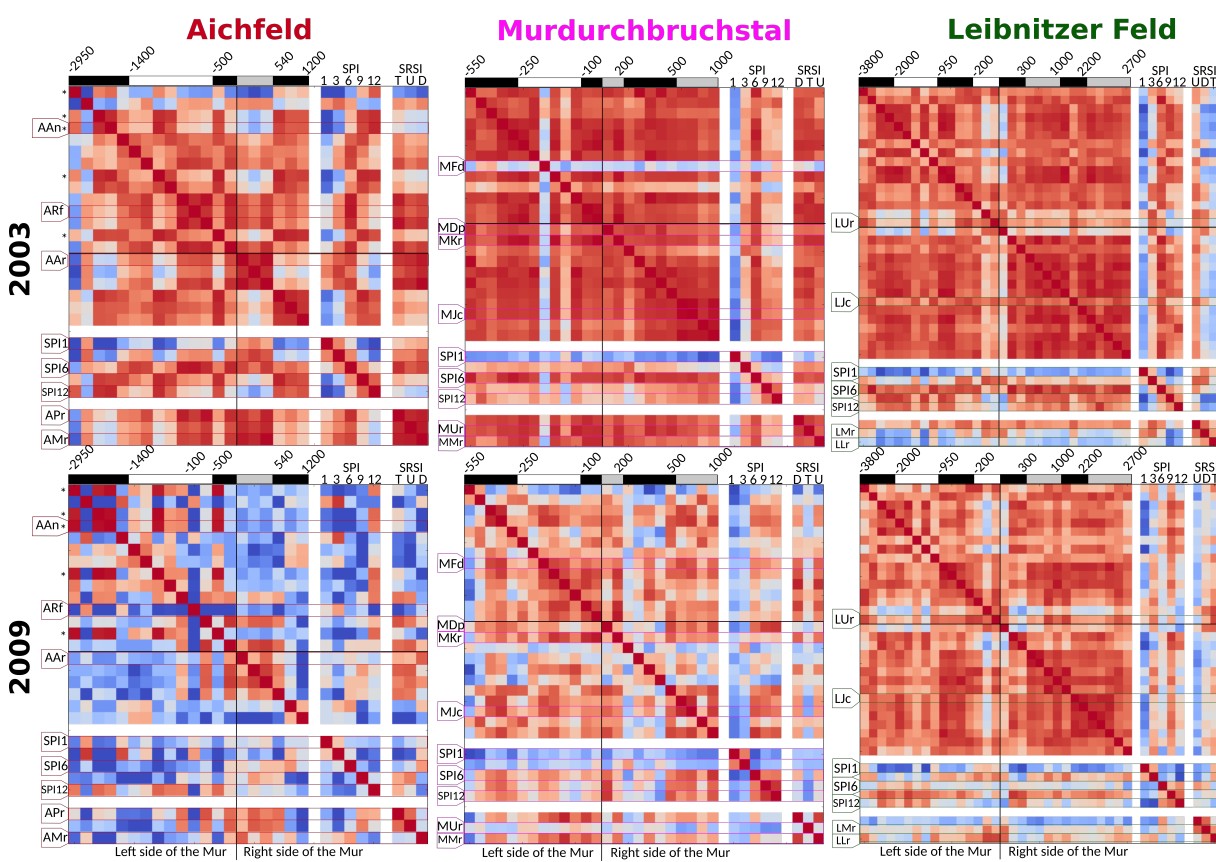

**Figure 3.** Correlation matrices for the three subregions, showing the effects of the drought year 2003 and the flood year 2009. Legend for the colors and description of the distances: see Figure 2

## 3.2 Selected flood and drought years

Figure 3 shows the correlation matrices for the standardized time series for the well known European drought year 2003 (see for example Beniston and Diaz (2004); van der Schrier et al. (2007); García-Herrera et al. (2010) and Nobilis and Godina (2006) and BMLFUW(2006) for Austria) and the local flood year 2009 (see BMLFUW(2011) for Austria and Hornich (2009);

5    Schatzl (2009); Stromberger et al. (2009) and Ruch et al. (2010) for the Mur region).

In the 2003 drought year, the Mur catchment saw only 80% of the 1961 - 90 average precipitation, 64% of discharge at the Mur in Leoben (between the Aichfeld and Murdurchbruchstal), 59% of discharge at the Mur in Spielfeld (downstream of the Leibnitzer Feld), compared with the 1991 - 2000 average and a general reduction in groundwater levels (BMLFUW, 2006). In the 2009 flood year, the Mur catchment saw 123% of the 1961 - 90 average precipitation, 128% of discharge at the

10   Mur in Leoben, 135% of discharge at the Mur in Spielfeld, compared with the 1991 - 2000 average and a general increase in groundwater levels (BMLFUW, 2011).

Compared with the correlations over the total time period (see Figure 2 and Table 3), the drought year generally, apart from the deep wells within the Aichfeld shows mostly highly significant ($p < 0.01$) higher correlations of the SGI time series with each other, with the SPI6 time series and with the SRSI time series and higher correlations between SRSI and SPI time series, albeit with differing significances. The flood year shows mostly highly significant ($p < 0.01$) lower correlations than the drought year. Compared with the total time period, the difference is not as visible as with the drought which is also visible in the somewhat reduced significances. The strongest difference between flood and drought is visible in the Aichfeld, where negative correlation prevails under flood conditions, going even lower than the -0.45 threshold chosen for the color scheme in the figures.

Another noticeable phenomenon is that certain wells can show a behavior that strongly deviates from their average behavior and the general trends for a given time span. For example well MFd in the Murdurchbruchstal and well ARf in the Aichfeld are among the wells with highly correlated SGI time series in their respective subregions for the complete time period (see Figure 2) but show low correlations under flood conditions (well ARf, 2009, Figure 3) or drought conditions (well MFd, 2003, Figure 3). Well MFd shows less wet conditions in spring and is less affected by the 2003 summer drought than most other wells in the subregion. Well ARf shows a drier spring and winter than most other wells in the subregion during the wet year 2009.

## 3.3 Selected snow rich and snow poor years

In order to compare the effects of a snow rich and a snow poor year on the groundwater system, we selected the winters of 1985/86 (snow rich) and 1989/90 (snow poor). In 1985/86 (Nov 1985 - Oct 1986), the average snow height (including the summer months) was 11.98 cm in the Aichfeld, 9.4 cm in the Murdurchbruchstal and 6.2 cm in the Leibnitzer Feld, with cumulated fresh snow of 390 cm in the Aichfeld, 274 cm in the Murdurchbruchstal and 193 cm in the Leibnitzer Feld. In 1989/90 (Nov 1989 - Oct 1990), the average snow height was 0.32 cm in the Aichfeld, 0.11 cm in the Murdurchbruchstal and 0.04 cm in the Leibnitzer Feld, with cumulated fresh snow of 55 cm in the Aichfeld, 23 cm in the Murdurchbruchstal and 9.3 cm in the Leibnitzer Feld.

Compared with the correlations over the total time period (see Figure 2 and Table 4), the snow rich year generally shows higher correlations between the SGI time series with each other and the SGI and SRSI time series whereas the snow poor year shows lower correlations. Similar to the situation with flood and drought (see Section 3.2) most of the differences are highly significant ($p < 0.01$) or significant ($p < 0.05$), although there are some non-significant differences (see Table 4). Comparing the snow rich year with the snow poor year, all of the differences, except for the correlations of the SRSI with the SPI6 time series, are highly significant ($p < 0.01$).

In all cases, some patterns also visible in Figures 2 and 3 remain. The set of five deeper wells in the Aichfeld is almost always visible, but appears clearest for the years 1985/86, with a sixth well showing a similar behaviour under these conditions. The highly correlated clusters close to the river in the Aichfeld and the Murdurchbruchstal also prevail, as do the two clusters in the top left and the bottom right of the Leibnitzer Feld.

**Table 3.** Average Pearson correlation coefficients of a subregion for SGI time series with each other, SGI time series with the SPI6 and SGI time series with SRSI with their standard deviations for each subregion for the complete time series, during drought (2003) and flood (2009) conditions. Also shown is the p-value indicating the the statistical significance of the difference between the correlation coefficients between the full time period, drought and flood conditions.

| Location & type of data | Avg. corr. coeff.±stdev all-time | p-value 2003 - all-time | Avg. corr. coeff.±stdev 2003 | p-value 2003- 2009 | Avg. corr. coeff. ±stdev 2009 | p-value 2009 all-time |
|---|---|---|---|---|---|---|
| SGI, Aichfeld[†] | 0.59 ±0.15 | < 0.01 | 0.71 ±0.21 | < 0.01 | 0.13 ±0.43 | < 0.01 |
| SGI, Murdurchbruchstal | 0.55 ±0.27 | < 0.01 | 0.80 ±0.23 | < 0.01 | 0.50 ±0.30 | < 0.05 |
| SGI, Leibnitzer Feld | 0.73 ±0.14 | < 0.01 | 0.77 ±0.19 | < 0.01 | 0.70 ±0.20 | < 0.01 |
| SGI-SPI6, Aichfeld[†] | 0.57 ±0.11 | < 0.01 | 0.83 ±0.12 | < 0.01 | 0.14 ±0.30 | < 0.01 |
| SGI-SPI6, Murdurchbruchstal | 0.51 ±0.090 | < 0.01 | 0.84 ±0.19 | < 0.01 | 0.53 ±0.27 | < 0.01 |
| SGI-SPI6, Leibnitzer Feld | 0.72 ±0.069 | < 0.01 | 0.83 ±0.12 | < 0.01 | 0.28 ±0.15 | < 0.01 |
| SGI-SRSI, Aichfeld[†] | 0.52 ±0.16 | < 0.01 | 0.75 ±0.19 | < 0.01 | 0.20 ±0.50 | < 0.01 |
| SGI-SRSI, Murdurchbruchstal | 0.55 ±0.22 | < 0.01 | 0.81 ±0.16 | < 0.01 | 0.42 ±0.30 | < 0.01 |
| SGI-SRSI, Leibnitzer Feld | 0.32 ±0.18 | > 0.05 | 0.32 ±0.31 | > 0.05 | 0.27 ±0.27 | > 0.05 |
| SRSI-SPI6, Aichfeld[†] | 0.48±0.088 | < 0.01 | 0.88 ±0.051 | > 0.05 | 0.27 ±0.33 | > 0.05 |
| SRSI-SPI6, Murdurchbruchstal | 0.39 ±0.095 | < 0.05 | 0.85 ±0.020 | > 0.05 | 0.34 ±0.35 | > 0.05 |
| SRSI-SPI6, Leibnitzer Feld | 0.36 ±0.12 | > 0.05 | 0.43 ±0.24 | > 0.05 | 0.41 ±0.30 | > 0.05 |

[†] Only shallow wells

## 3.4 Development over time

Figure 5 and Table 5 show the development of the three subregions when split-up into time periods of 12 years (1975-1986, 1987-1998, 1999-2010). It should be noted that the Murdurchbruchstal only got a significant number of groundwater wells after 1980, so the first time period differs for this region, and is only 7 years long, from 1980 to 1986.

5     In the Aichfeld, there is no noticeable trend over time, besides the deviating behavior of the deep wells in the last period. As mentioned in Section 3.1.1, we are going to focus on analyzing the shallow wells in the Aichfeld. From the first to the second period, we see an increase in SGI correlations for a cluster of wells around the river and thus an increase of correlation of those wells SGI time series with the SRSI time series. In the last period, these correlations decrease. These small changes in the correlations of the SGI time series are also reflected by their average correlation coefficients (see Table 5). However the

10   averages do not necessarily reflect the significance of the change. While the second and last time period have similar averages, the change in the underlying set of SGI-SGI correlations is still significant (p < 0.05), as it is the case with the SGI-SPI6 correlations which also show a significant (p < 0.05) change from the second to the last period. On the other hand, the average

**Table 4.** Average Pearson correlation coefficients of a subregion for SGI time series with each other, SGI time series with the SPI6 and SGI time series with SRSI with their standard deviations for each subregion for the complete time series, during snow rich (1985/86) and snow poor (1989/90) conditions. Also shown is the p-value indicating the the statistical significance of the difference between the correlation coefficients between the full time period, snow rich and snow poor conditions.

| Location & type of data | Avg. corr. coeff.±stdev all-time | p-value 1985/86- all-time | Avg. corr. coeff.±stdev 1986/86 | p-value 1985/86- 1989/90 | Avg. corr. coeff. ±stdev 1989/90 | p-value 1989/90- all-time |
|---|---|---|---|---|---|---|
| SGI, Aichfeld[†] | 0.59 ±0.15 | < 0.01 | 0.79 ±0.19 | < 0.01 | 0.29 ±0.46 | < 0.01 |
| SGI, Murdurchbruchstal | 0.55 ±0.27 | < 0.01 | 0.79 ±0.16 | < 0.01 | 0.52 ±0.31 | > 0.05 |
| SGI, Leibnitzer Feld | 0.73 ±0.14 | > 0.05 | 0.73 ±0.26 | < 0.01 | 0.40 ±0.44 | < 0.01 |
| SGI-SPI6, Aichfeld[†] | 0.57 ±0.11 | < 0.05 | 0.72 ±0.21 | < 0.01 | 0.22 ±0.34 | < 0.01 |
| SGI-SPI6, Murdurchbruchstal | 0.51 ±0.090 | < 0.01 | 0.77 ±0.10 | < 0.01 | 0.026 ±0.31 | < 0.01 |
| SGI-SPI6, Leibnitzer Feld | 0.72 ±0.069 | < 0.05 | 0.80 ±0.21 | < 0.01 | 0.45 ±0.27 | < 0.01 |
| SGI-SRSI, Aichfeld[†] | 0.52 ±0.16 | < 0.01 | 0.74 ±0.16 | < 0.01 | 0.33 ±0.50 | < 0.05 |
| SGI-SRSI, Murdurchbruchstal | 0.55 ±0.22 | < 0.01 | 0.67 ±0.18 | < 0.01 | 0.44 ±0.32 | < 0.05 |
| SGI-SRSI, Leibnitzer Feld | 0.32 ±0.18 | < 0.01 | 0.60 ±0.16 | < 0.01 | -0.055±0.41 | < 0.01 |
| SRSI-SPI6, Aichfeld[†] | 0.48 ±0.088 | > 0.05 | 0.62 ±0.14 | > 0.05 | -0.24 ±0.23 | < 0.05 |
| SRSI-SPI6, Murdurchbruchstal | 0.39 ±0.095 | < 0.05 | 0.72 ±0.042 | > 0.05 | -0.18 ±0.41 | > 0.05 |
| SRSI-SPI6, Leibnitzer Feld | 0.36 ±0.12 | < 0.05 | 0.66 ±0.099 | > 0.05 | -0.14 ±0.44 | > 0.05 |

[†] Only shallow wells

correlation coefficients can show noticeable changes between the time periods, but the changes are not significant ($p > 0.05$), as it is the case for the SGI-SRSI correlations.

The Murdurchbruchstal shows similar behavior in the first and second period, with some slightly different clusters. In the first period, the upstream and downstream Mur gauges show SRSI time series highly correlated with each other. In the last period, we see higher correlations of all SGI time series with each other, the SPI time series and the SRSI time series, with only the one month SPI and the downstream Mur gauge showing some low correlations. These visible changes are also reflected in the average correlation coefficients for the SGI time series within the subregion (see Table 5). Highly significant ($p < 0.01$) changes occur in the SGI-SGI and SGI-SPI6 correlations between the second and last period, as well as for all periods for the SGI-SRSI correlations.

The Leibnitzer Feld also shows a slight decrease in correlations in the middle period, followed by a strong increase in the last time period. Compared with the complete time period shown in Figure 2, the Leibnitzer Feld shows higher correlations of SGI time series with the SRSI time series for the shorter time periods, but wells close to the river show a comparably lower correlated SGI time series. The mentioned decrease followed by an increase is reflected by highly significant ($p < 0.01$) changes of the correlation coefficients for the SGI time series within the subregions for the first, second, and third time period.

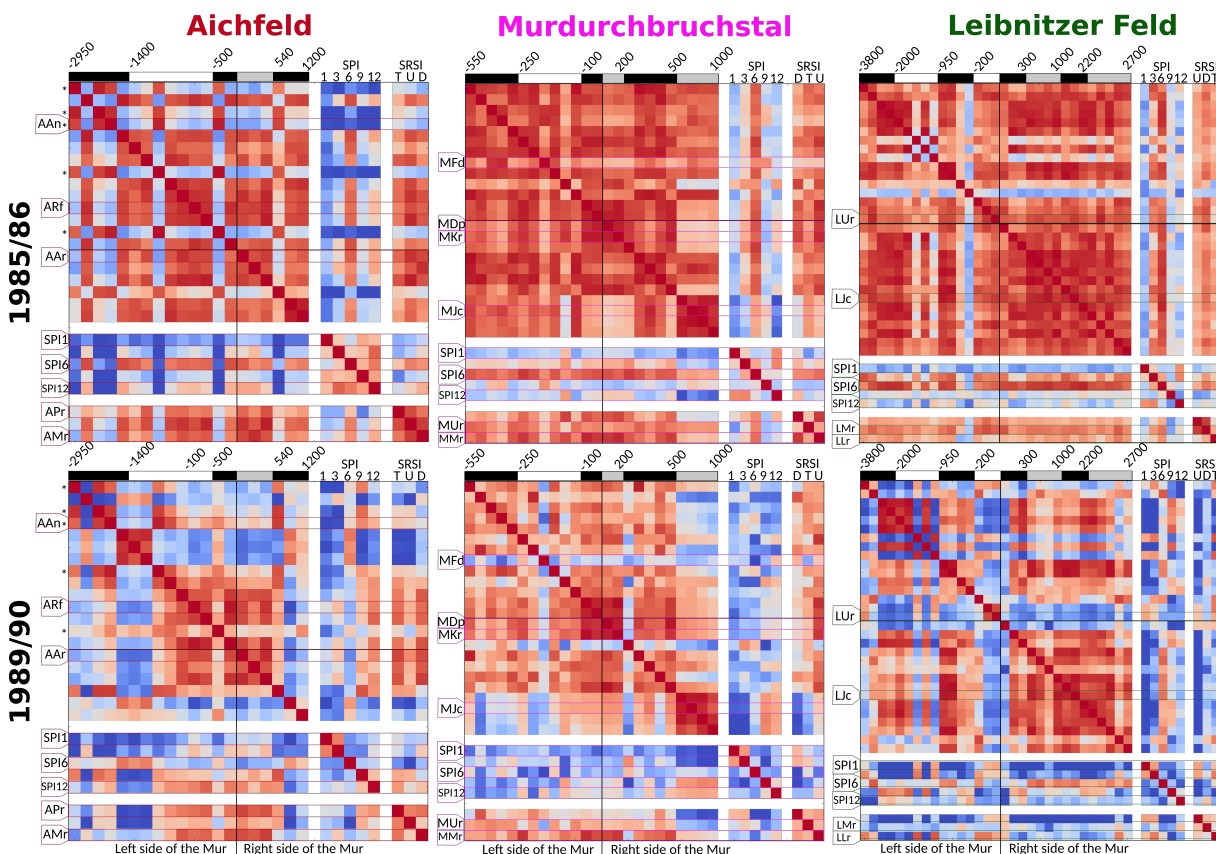

**Figure 4.** Correlation matrices for the three subregions, showing the effects of the snow rich winter of 1885/86 and the snow poor winter of 1989/90. Legend for the colors and description of the distances: see Figure 2

The correlations of SGI and SPI6 also seem to follow the decrease-increase pattern, with highly significant (p < 0.01) changes between all three periods. Only the SGI-SRSI correlations deviate from the general pattern and show no significant (p > 0.05) change between the first and the second period.

# 4  Discussion

## 4.1  Spatial variability

As already shown in Section 3.1, a large number of groundwater wells in each subregion shows SGI time series highly correlated with each other. Some of those wells are also in close vicinity to each other (e.g. the cluster of highly correlated wells in the Murdurchbruchstal subregion, all located in the town Gratwein-Straßengel), to the river Mur (e.g. most of the shallow wells in the Aichfeld subregion) or located in a similar geologic setting (e.g. the deep wells in the Aichfeld subregion or almost all the shallow wells in the Leibnitzer Feld subregion).

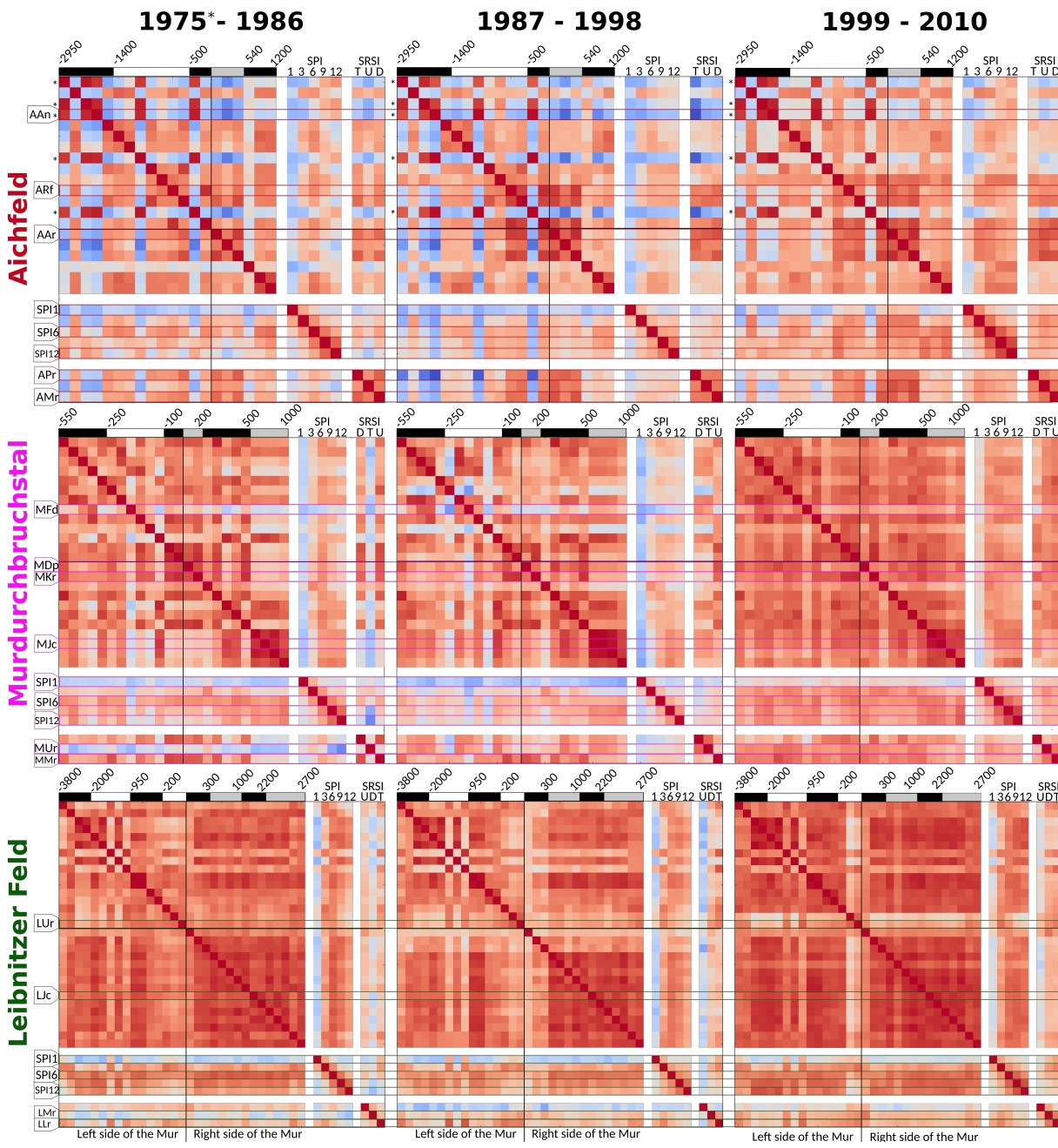

**Figure 5.** Correlation matrices for the three subregions split into three time periods. Note that the first period for the Murdurchbruchstal is from 1980 - 1986 due to lack of data before 1980. Legend for the colors and description of the distances: see Figure 2

**Table 5.** Average Pearson correlation coefficients of a subregion for SGI time series with each other, SGI time series with the SPI6 and SGI time series with SRSI with their standard deviations for each subregion and each time period. Also shown is the p-value indicating the the statistical significance of the difference between the correlation coeffcients of the two time periods.

| Location & type of data | Avg. corr. coeff. ±stdev 1975[*] - 1986 | p-value | Avg. corr. coeff. ±stdev 1987 - 1998 | p-value | Avg. corr. coeff. ±stdev 1999 - 2010 |
|---|---|---|---|---|---|
| SGI, Aichfeld[†] | 0.63 ±0.16 | > 0.05 | 0.62 ±0.15 | < 0.05 | 0.66 ±0.11 |
| SGI, Murdurchbruchstal | 0.64 ±0.19 | > 0.05 | 0.62 ±0.20 | < 0.01 | 0.75 ±0.10 |
| SGI, Leibnitzer Feld | 0.77 ±0.12 | < 0.01 | 0.71 ±0.15 | < 0.01 | 0.80 ±0.14 |
| SGI-SPI6, Aichfeld[†] | 0.62 ±0.14 | > 0.05 | 0.55 ±0.075 | < 0.05 | 0.67 ±0.072 |
| SGI-SPI6, Murdurchbruchstal | 0.61 ±0.12 | > 0.05 | 0.48 ±0.13 | < 0.01 | 0.67 ±0.057 |
| SGI-SPI6, Leibnitzer Feld | 0.76 ±0.069 | < 0.01 | 0.69 ±0.082 | < 0.01 | 0.74 ±0.060 |
| SGI-SRSI, Aichfeld[†] | 0.58 ±0.13 | > 0.05 | 0.55 ±0.19 | > 0.05 | 0.58 ±0.17 |
| SGI-SRSI, Murdurchbruchstal | 0.44 ±0.26 | < 0.01 | 0.56 ±0.17 | < 0.01 | 0.65 ±0.11 |
| SGI-SRSI, Leibnitzer Feld | 0.41 ±0.15 | > 0.05 | 0.38 ±0.21 | < 0.01 | 0.46 ±0.15 |

[*] For the Murdurchbruchstal this period is from 1980 - 1986 due to lack of data before 1980; [†] Only shallow wells

As a result of the different behavior of the groundwater wells in the different subregions, the correlations of SGI time series with the SPI time series also differ between the subregions (see Table 2). While the SPI1 still shows similar, low average correlations, the longer SPI averaging periods show a different behavior in the subregions. Hence, we are only discussing the higher averaging periods, except for parts of the Aichfeld.

5   Since there are 2 distinct aquifer bodies in the Aichfeld, the groundwater data was split-up into a shallow (average depth of the wells: 9.7 m) and a deep (average depth of the wells: 24.9 m) part. The deep wells SGI time series are only lowly correlated with SPI time series, with a minimum for the SPI1 of -0.04 and a maximum of 0.38 for the SPI12. The average SPI-SGI correlations for the shallow wells range from 0.38 for the SPI12, to a maximum of 0.57 for the SPI6.

In the Murdurchbruchstal, where all of the wells have similar depths (average: 10.7 m), the average correlations between 10   SGI and SPI time series range from a minimum of 0.41 for the SPI3 to a maximum of 0.51 for the SPI6.

The wells in the Leibnitzer Feld also have similar depths (average: 6.4 m). Here, the average correlations between SGI and SPI time series range from 0.58 for the SPI3 to 0.72 for the SPI6.

All subregions (or the shallow part of the subregion in the case of the Aichfeld) have the highest correlation with the SPI time series for an averaging period of 6 months. Only the deep part of the Aichfeld has its maximum correlation with the 12 month 15   SPI, which fits the findings of Kumar et al. (2016) who found that deeper wells correlate better with longer SPI averaging periods.

The SPI6 - SGI correlations follow the average depths of the wells, with the highest correlation found in the most shallow Leibnitzer Feld, and the lowest correlation found in the deep part of the Aichfeld, a pattern that is also repeated for all other

averaging periods. The shallow part of the Aichfeld and the Murdurchbruchstal have very similar average depths (9.7 and 10.7 m, compared to 24.9 m for the deep Aichfeld and 6.4 m for the Leibnitzer Feld), so that they show similar correlations, ranging between those of the deep wells in the Aichfeld and the shallow wells in the Leibnitzer Feld.

In all regions, there is a low correlation between standardized river stages and standardized precipitation, with an average correlation coefficient for SPI with SRSI ranging from 0.47 in the Aichfeld to 0.37 in the Leibnitzer Feld (see also Table 2), with the highest correlations between river and precipitation generally found for the 3 and 6 month SPI. This suggests that, in addition to the transformation of the rainfall signal due to the runoff processes within the subregion, the rivers can transport a precipitation signal from a region upstream of the subregion in question, which can have a different precipitation signal from the local precipitation. This is also supported by the fact that the differences between the correlations of the SRSI time series from each subregion with the SRSI time series from the different subregions appear not to be significant ($p > 0.05$). Also, this upstream signal can in itself be a "collection" of many different regional precipitation patterns. This suggests that the correlation of the SRSI time series with the 3 and 6 month SPI results from the influence of the large, general "climate" in the region.

Another factor affecting the rivers are the numerous run-of-the-river power plants, which alter the natural course and timing of the rivers and remove their natural short-term precipitation signal. For the Aichfeld, where there are only 5 small-scale power plants in its upstream part, this does not affect the river Mur too much, shown by the high average correlation of the river gauging stations SRSI time series with each other of 0.65. A similar value of 0.61 is observed in the Murdurchbruchstal, even though there are 8 hydro power plants in the subregion. In the Leibnitzer Feld however, the combination of 5 power plants, and the fact that the gauging stations are located outside of the subregion results in an average correlation of the SRSI time series with each other of only 0.17. However, as mentioned above, the differences in SRSI time series between the subregions are still not significant ($p > 0.05$).

Thus in small systems such as the Aichfeld and the Murdurchbruchstal - and to some amount probably also the Leibnitzer Feld -, the river and the groundwater will be closely related to each other. At high water levels, the river feeds the groundwater, thus superpositioning its signal onto the groundwater, whereas the groundwater provides the river baseflow in low water conditions, thus controlling river flow and river stage at low water levels (see also Section 4.2).

In summary, the most obvious differences between the subregions are the low correlation of the river gauges SRSI time series with the groundwaters SGI time series in the Leibnitzer Feld, described in detail in Section 3.1.3, and the differences between SGI-SPI correlations, where Aichfeld and Murdurchbruchstal show generally low to moderate correlations, and the Leibnitzer Feld shows generally high to very high correlations, following the thickness of the aquifers in the subregions.

## 4.2 Selected flood and drought years

As shown in Figure 3 and Section 3.2, the drought and flood years of 2003 and 2009 show a very different behavior in the regions investigated herein. As mentioned in Section 3.1.1, we are mainly going to discuss the shallow Aichfeld, since the shallow aquifer is directly affected by this relatively short term events.

Generally, we see an increase in correlations under drought conditions and a decrease under flood conditions, which is not only reflected by the color coded, single correlations coefficients shown in Figure 3.2 but also by most average correlations coefficients shown in Table 3. Apart from the correlations between the SGI and SRSI time series in the Leibnitzer Feld and all of the correlations between the SRSI and SPI6 time series which do not show a significant change ($p > 0.05$), all of the differences between the 2003 drought year and the 2009 flood year are highly significant ($p < 0.01$).

In order to interpret these differences, it is important to look at the differences in the underlying drought and flood. As shown in Section 3.2, the 2003 drought was a long term and large-scale event, affecting all of Europe for most of the year (e.g. Beniston and Diaz (2004); Nobilis and Godina (2006); van der Schrier et al. (2007); García-Herrera et al. (2010) and BMLFUW(2006)). The 2009 flood on the other hand, was a more small-scale event, split-up into multiple flood peaks (e.g. Hornich (2009); Schatzl (2009); Stromberger et al. (2009); Ruch et al. (2010) and BMLFUW(2011))

The 2003 deficit of only 59% of discharge at the Mur gauge in Spielfeld (BMLFUW, 2006) was the result of long term and country wide dry conditions, whereas the 2009 excess discharge of 135% in Spielfeld (BMLFUW, 2011) is the result of multiple flood events, often very localized in the small tributaries to the Mur (Schatzl, 2009; Hornich, 2009), partly also resulting in considerable, localized overbank flow. While the 2003 drought showed a slow decrease in water levels in the aquifer and the rivers, the 2009 flood showed fast increases in water levels, which in case of the rivers get transported downstream to an area that might not be affected by a localized precipitation maximum.

The observation that a long-term drought affects the whole aquifer and that a short-term flood only affects parts of the aquifer fits the idea of aquifer response times (Downing et al., 1974; Alley et al., 2002). The aquifer response time $T^*$ is a function of storativity ($S$), "some characteristic length" of the aquifer ($L$) and the transmissivity ($T$): $T^* = \frac{S \times L^2}{T}$. We approximate $S$ for our unconfined case by the specific yield ($S_y$) and $T$ by multiplying the average K of a subregion with its average saturated aquifer thickness (see Sections 2.1.1, 2.1.2 and 2.1.3). For $S_y$ we use a value close to the average porosity of 22% compiled by Fank et al. (1993). For $L$ we are using the average distance perpendicular to the river Mur within which most wells of a particular subregion are situated. With these values and assumptions, we obtain values for $T^*$ ranging from over 1 month to over 1 year. Thus a short event, such as a flood will not affect the whole aquifer, whereas a long term event such as the 2003 drought affects the whole area or at least most parts of it. The aquifer response time also offers a possible interpretation of the deeper aquifer in the Aichfeld, which generally shows high correlations of SGI time series with each other, irrespective of conditions, but especially so under flood conditions (see Figure 3). The deep aquifer likely is confined or semi-confined, so that the storativity S is orders of magnitude lower than the $S_y$ of the shallow unconfined aquifers and thus results in response times from hours to days. This allows for all of the wells in question to react to a perturbation such as a short flood well within the one month time scale shown in the correlation matrices.

The phenomena discussed above also match the findings of Eltahir and Yeh (1999), who stated that droughts have a much more "persistent signature on groundwater hydrology , in comparison to [...] floods". They suggest that floods - increases in groundwater levels - can dissipate very quickly by groundwater discharge, whereas there is no dissipation mechanism available for low groundwater levels. Following this interpretation, Eltahir and Yeh (1999) argue that this explains the asymmetry of the water levels response to a flood or drought event and suggest that this mechanism deserves further investigation. We argue that

this asymmetry is not only seen in a single hydrograph, but also in the whole area, resulting in the different pictures shown in Figure 3, where only the SPI1 shows similar correlations under flood and drought conditions.

Looking at the parts of the aquifers not influenced by rivers, an increase in precipitation will increase infiltration and thus simply increase the water levels, keeping the general flow direction and thus correlations between neighboring wells time series

intact, shown by the areas of high correlations in Figure 3. However, looking at the parts of the aquifer close to the river - which includes many wells that are close to small creeks and streams that are not considered for the general discussion in this paper - a multitude of possible phenomena is seen. As a direct pathway, bedload during floods can erode the clogging layer in the river bed and thus provide a significant short-time improvement in infiltration (Schubert, 2002). Sophocleous (1991) shows that river floods can transport pressure pulses in highly conducting channels, as described in Zetinigg et al. (1966). A similar

phenomenon, is shown by Vekerdy and Meijerink (1998) following floods through the aquifer for distances of over 2 km. Doble et al. (2012) describes wells at similar distances that show a strong and fast reaction to a river flood within 1.5 to 6 days, both with inundation and without. In a further paper, Doble et al. (2014) argue that "overbank flood recharge is not an insignificant volume". As discussed in Workman and Serrano (1999), flood events - with overbank flow - can make up significant parts of the recharge in river-close parts of an aquifer.

The mechanisms described above can result in two phenomena besides the still existing baseflow: a pressure pulse propagating through the aquifer or a real and rapid infiltration, both being oriented against the usually dominating flow towards the river, and a potential for local backwaters where the inflow from the river and the baseflow towards the river meet. This results in similar changes in all of the aquifer under normal and drought conditions, resulting in high correlations, whereas flood conditions can cause differing changes in the aquifer, resulting in low correlations.

**4.3   Selected snow rich and snow poor years**

As shown in Figure 4 and Section 3.3, the snow rich and snow poor years of 1985/86 and 1989/90 show a very different behavior in the regions investigated herein. As mentioned in Section 3.1.1, we are mainly going to discuss the shallow Aichfeld, since the shallow aquifer is directly affected by this relatively short term events.

Generally, we see an increase in correlations under conditions with a lot of snow and a decrease under conditions lacking

snow, which is not only reflected by the color coded single correlation coefficients shown in Figure 4 but also by most average correlation coefficients shown in Table 4. The differences between the snow rich year 1985/86 and the snow poor year 1989/90 are all highly significant ($p < 0.01$), except for the differences between the correlations of the SRSI time series with the SPI6 time series in all subregions ($p > 0.05$). As with drought and flood, we have again singled out the SPI6 for detailed investigation, since this appears to be the highest correlated SPI averaging period. Unlike drought and flood however, the SPI6 is the only SPI

averaging period that shows consistently high correlations (compare Figures 3 and 4) under snow rich conditions. In contrast to SPI1 and SPI3, SPI6 is highly correlated with SGI in the snow-rich year, suggesting that an aggregation period of 6 months is sufficient to account for the effect of the snow accumulation, which prohibits most groundwater recharge, just as a lack of precipitation under drought conditions does. However while drought conditions still allow for a connection of precipitation and groundwater, a closed snow cover essentially breaks this connection, with subsequent precipitation just adding to the existing

snow cover. It is noteworthy that the correlations are also much weaker though for the longer aggregation periods (SPI9 and SPI12) just as generally observed in the entire time series of all three subregions (see Section 3.1).

The observation that a snow rich year affects the whole aquifer whereas a snow poor year affects parts of the aquifer also fits the idea of aquifer response times as discussed in Section 4.2. Since the aquifer response time $T^*$ ranges from over 1 month to over 1 year, a lack of snow will enable the aquifer to react to short term and localized events such as precipitation, melt or flood events, whereas the delayed groundwater recharge and runoff under snow rich conditions is a long-term event that will be able to affect the whole aquifer.

## 4.4 Development over time

As shown in Figure 5 and Section 3.4, the Murdurchbruchstal and Leibnitzer Feld subregions show an increase of correlations with time within the aquifer and between the aquifer and the rivers and the precipitation time series. In contrast, the Aichfeld shows no clear trend over time. As shown in table 5, this is also in part reflected by the significances of the changes between the periods. While the Murdurchbruchstal and the Leibnitzer Feld show highly significant changes from the second to the last period, the shallow Aichfeld does not. Also the changes from the first to the second period are in part significant or highly significant for the Murdurchbruchstal and the Leibnitzer Feld, whereas the Aichfeld only shows insignificant changes for these periods.

Compared with the increased correlations under drought conditions (Sections 3.2 and 4.2), one simply could assume that the split-up time series show a development towards dryer conditions, which is in line with the general assumption of an already warming and drying climate for Austria (Kromp-Kolb et al., 2014). Another assumption could be that the split-up time series show a development towards increasing amounts of snow, as the comparison of snow rich and snow poor years has shown that the correlations between the SGI time series of the wells and those between the SGI and the SPI tends to be stronger in the snow rich years. However, looking at the underlying means (see Figures 6, 7, 8 and 9), a different picture manifests itself. The average unstandardized snow levels and fresh snow amounts shown in Figure 9 show an increase in snowfall and heights in the first period, with a sharp drop, followed by a strong increase again in the second period and a drop and an unsteady development in the third period. Thus, the most recent time period, which exhibits the highest correlations, is clearly less affected by snow than the preceding time periods. This is contrary to the observation made when comparing snow-rich and snow-poor years (Section 4.3). Thus, the following discussion focuses on a tendency towards drier conditions as potential explanation for the observed increase in correlations.

While the average SPI in all regions remains more or less stable, there are some noticeable changes in SGI and SRSI. As shown in Figure 6, the Aichfeld shows a slight increase in groundwater levels for the first half of the time, followed by a decrease, whereas the Murdurchbruchstal (see Figure 7) shows an increase in groundwater and river water levels in all time periods. Contrary to those two regions, the Leibnitzer Feld, shown in Figure 8, shows an incoherent signal.

When analyzing the occurrence of extreme events (SGI, SPI and SRSI below/above -2/+2), we observe the following:

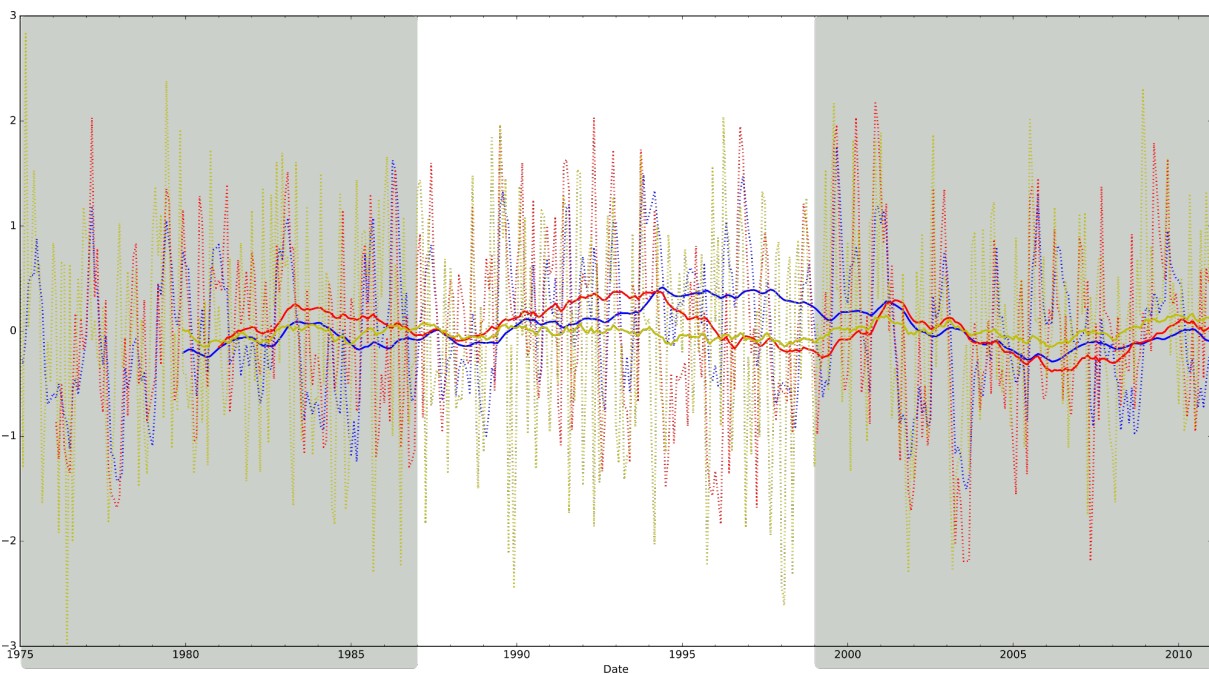

**Figure 6.** Average values (dotted lines) for the SGI (blue), SPI1 (yellow) and SRSI (red) and their 5 year running means (solid lines) for the Aichfeld subregion.

For values below -2, the SPI1 has the largest count in 1987 - 1998 in the Aichfeld and the Murdurchbruchstal and in 1987 - 1998 and 1999 - 2010 in the Leibnitzer Feld. This only is reflected in the groundwater in the Leibnitzer Feld, where the largest count of below -2 events is seen in the 1999 - 2010 SGI.

The SGI does not follow this pattern for the Aichfeld and the Murdurchbruchstal, where the highest count is observed in the 1975 (1980) - 1986 period, medium count is observed in 1999 - 2010 and lowest count is observed in 1987 - 1998. As shown in Sections 3.1 and 4.1, the SPI6 is highest correlated to the SGI. For this SPI averaging period, the highest count of below -2 events is observed in the 1999 - 2010 period in all subregions.

Only the Leibnitzer Feld shows the highest number of below -2 events in the same (1999 - 2010) period in the SGI, the SPI1 and SPI6, which is another indicator for the dominant role of precipitation in this subregion.

The most extreme values below -2.5 only occur in SPI, most prominently in the Murdurchbruchstal, where we are observing an increase from 0 in the 1980 - 1986 period to 2 events (one each in SPI1 and SPI3) in 1987 - 1998 to 17 (SPI6: 3; SPI9: 6; SPI12: 8) in 1999 - 2010. The other subregions show smaller counts, with most of the below -2.5 events being observed in the higher SPI averaging periods and the 1999 - 2010 period.

The SRSI behaves inconclusive. For the Aichfeld it shows the same pattern of events with values below/above -2/+2 as the SPI6, indicating a delayed precipitation controlled river system. In the Murdurchbruchstal, it follows the same pattern as the groundwater, which fits the interpretation of the rivers being the driver of the groundwater dynamics.

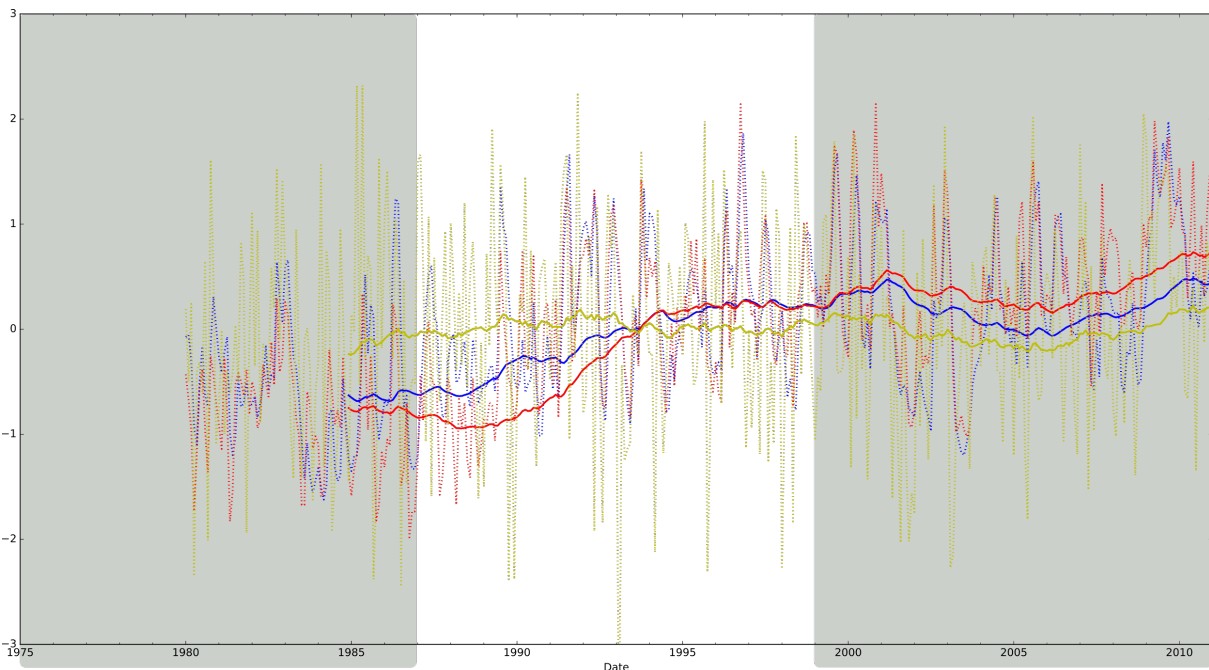

**Figure 7.** Average values (dotted lines) for the SGI (blue), SPI1 (yellow) and SRSI (red) and their 5 year running means (solid lines) for the Murdurchbruchstal subregion.

The SRSI pattern of the Murdurchbruchstal (highest counts of negative events in 1975/1980 - 1986, lowest in 1987 - 1998) is also seen in the Leibnitzer Feld, but here it fits neither the behavior of the SPI nor that of the SGI. This is in accordance with our other observations that the river in this subregion is intensively human influenced and that both, the upstream and the downstream gauging station are outside of the subregion.

For extreme flood values above +2.5, it is again only the SPI where those occur, but with a lower count of only 1 in SPI1 and SPI3 each in the Aichfeld in the 1975 - 1986 period and 11 (SPI6: 1; SPI9: 4; SPI12: 6) in the Murdurchbruchstal in the 1999 - 2010 period.

SPI1 and SPI6 values above +2 show the same patterns as SPI1 and SPI6 values below -2, with the largest counts mostly occurring in the 1999 - 2010 period. In contrast SGI above +2 shows the 1975 - 1986 period as the wettest in the Aichfeld and

the Leibnitzer Feld. Only in the Murdurchbruchstal, the highest count of +2 SGI events occurs in the time period from 1999 - 2010.

The SRSI also shows inconsistent patterns for positive events, where only the Murdurchbruchstal has the same behavior in the +2 SRSI as it does in the +2 SGI, confirming again the influence of the river on the groundwater.

This different patterns follow our previous interpretation of river dominated upstream subregions and a precipitation domi-

nated Leibnitzer Feld. It thus appears that the influence of precipitation is sufficient to cause a similar behavior in groundwater levels within the shallow aquifer of the Leibnitzer Feld, while it is overruled by direct human impacts in the upstream part of

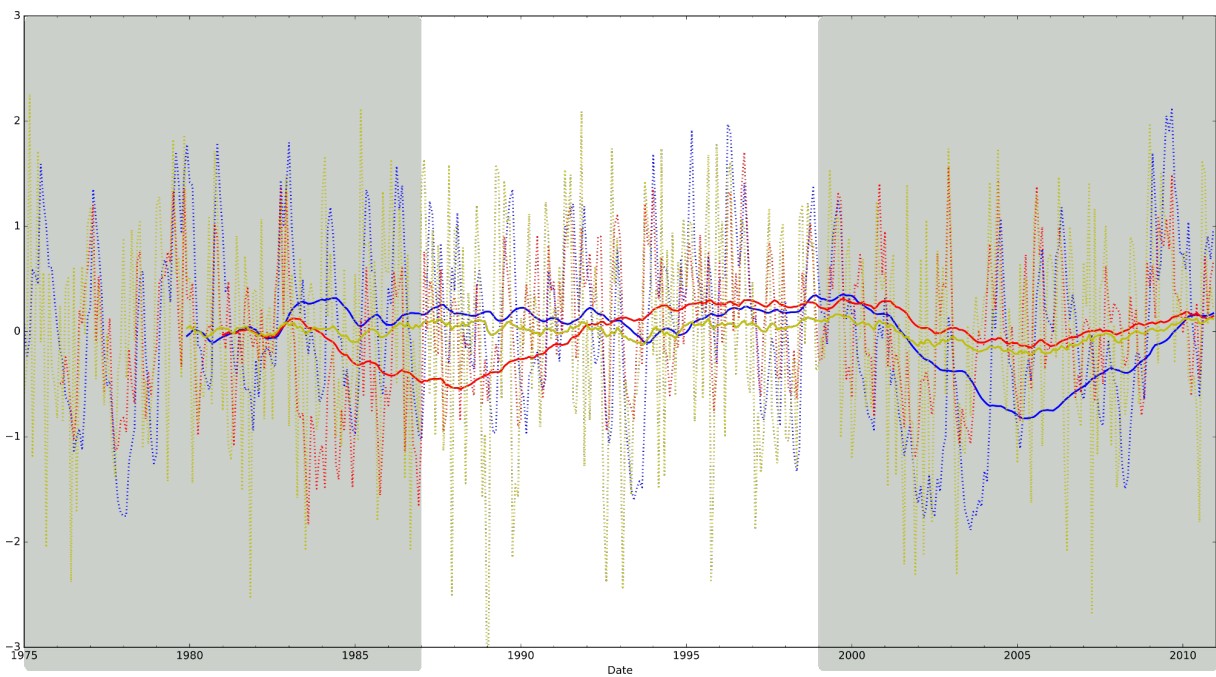

**Figure 8.** Average values (dotted lines) for the SGI (blue), SPI1 (yellow) and SRSI (red) and their 5 year running means (solid lines) for the Leibnitzer Feld subregion.

the catchment. When looking at detailed time series (e.g. well MJc or river gauge MUr in Figure 2), it becomes obvious that many events above the +/-2 threshold are not flood or drought events, but result from an overlying trend or are the result of direct human activities. The only exception from this is the SPI since there is no direct human influence on precipitation. This poses the question of the feasibility of the indices, which is going to be discussed in the following section.

## 4.5 Feasibility of the indices and synthesis

As already discussed in Section 2.2, SPI and - to a smaller amount - SGI have seen considerable use. However, the shallow aspect of most of our region presents a challenge to the SGI - or similar indices such as the SRSI: It has been suggested that the assumption of stationarity underlying many hydrogeological and hydrological assessments and the engineering decisions based upon them is inadequate in view of the ongoing hydroclimatic change (Milly et al., 2008; Koutsoyiannis, 2010, 2011). In fact, some of the time series singled out in this investigation show a behavior that is nonstationary within the observed period of time. Besides the looming threat of climate change, as for example mentioned by Milly et al. (2008), various events that cause a deviation from a stationary trajectory (see also Section 3.4 and Section 4.4) can be observed. As shown in Figure 2, the wells MDp, MKr and MJc and the river gauges MUr and LLr exhibit a split pattern, where at a certain point in time, the standardized values change from a wet( dry) dominated to a dry( wet) dominated regime.

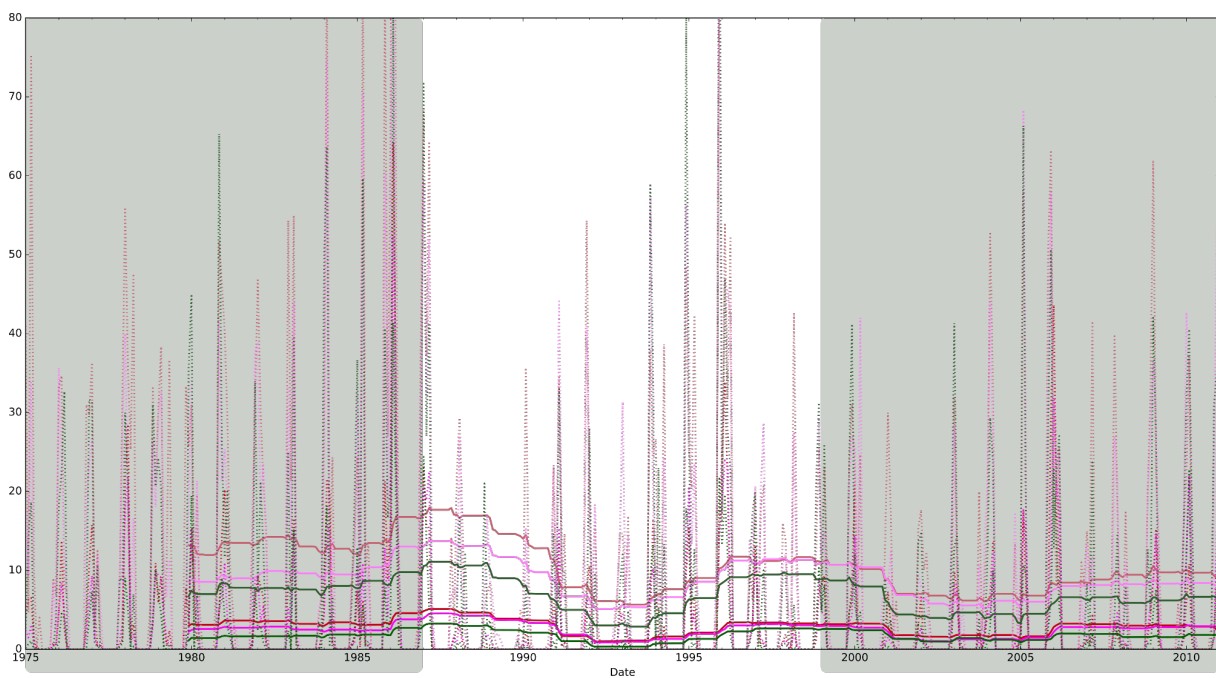

**Figure 9.** Average values (dotted lines) and their 5 year running means (solid lines) for the fresh snow (pale colors) and the snow height for the Aichfeld subregion (red), the Murdurchbruchstal subregion (purple) and the Leibnitzer Feld subregion (green).

For some of the time series in question the reason can be easily found. In the case of MDp and MKr, it is the construction of the power plant "Friesach" in 1998 with a pondage of approx. 7 m upstream and a decrease in tailwater of approx. 1 m (VERBUND AG, 2016a). Well MDp is situated approx. 200 m upstream of the weir, and thus shows "dry" conditions before the construction and "wet" ones afterward. MKr is just located 1.1 km downstream of MDp and 1 km downstream of the weir,

and thus shows "wet" conditions before the construction and "dry" ones afterward.

Other time series also seem to be linked to a certain event, such as the case with MJc and MUr, where a change from a wet to a dry regime happens around 1990. However, in this case, both points are situated 9 km apart from each other, and none of the power plants that could affect them have been built at the time in question.

It is interesting to note that those time series discussed above are very similar to the synthetic time series discussed in

Koutsoyiannis (2011), most notably Figure 3 in the cited paper. Koutsoyiannis (2011) discusses a synthetic time series that is running for 1000 terms which has the following properties: When looking at the first 50 terms, it appears very irregular but can be assumed to have a constant mean over time. We argue that this phenomenon is also visible in our time series MDp and MKr, where the period from 1975 to 1998 shows a similar behavior to Koutsoyiannis synthetic series. Zooming out, Koutsoyiannis (2011) shows the first 100 terms of the synthetic time series, which now show two distinctive periods with different averages and an apparent "shift" or "change" between those periods. This phenomenon is also visible in our time series MDp and MKr,

where this apparent "shift" or "change" occurs around 1998. Following this, Koutsoyiannis (2011) zooms out further to 1000

terms of the synthetic time series, to show that it still is stationary in the long term. This latter step cannot be seen in our data, but we would expect a similar picture, when looking at the time series MDp and MKr 1000 years from now. This effect of apparent stationarity when zooming in can also be seen in Sections 3.4 and 4.4, where it becomes apparent that the split-up time series are generally showing higher correlations than the full time series, since only comparably smaller parts of the time

periods are affected by a large change.

A quantification and counting of extreme events for the full time, as attempted in Section 4.4, is thus problematic. Calling e.g. an index value of -1 to -1.49 "moderate drought" (McKee et al., 1993) can be misleading when assessing a nonstationary time series, such as well MDp (Figure 2). Here the first approx. 18 years would be interpreted as a period of multiple and persistent moderate to severe droughts, followed by a period of multiple and persistent moderate to severe floods. What the

time series really shows is an aquifer in equilibrium with its surroundings before and after the construction of a run-of-the-river power plant and the associated change in groundwater level. To enable a quantification of the negative (and positive) events, the time series in question could be split-up at the time of the change, standardized independently and put back together. However, this requires knowledge of the nature and the timing of the underlying events, which in our case was not always available.

For systems understanding and correlation however, these jumps in time series are not an issue. As shown with wells MDp

and MKr, the construction of a run-of-the-river power plant does not only change the water levels of the river in question, it also does affect the groundwater up- and downstream of it. With our matrix-view (Figure 2), it can be shown that this change did not only affect the two wells singled out, but also at least one other well downstream in the case of MKr, where the first "blue outlier" above it is situated directly across the Mur from MKr. The second one however is upstream of MKr and its power plant, but in a similar downstream distance from another power plant (VERBUND AG, 2016b). With well MDp, there are at

least two other wells upstream that show very high correlations with MDp.

This shows that large events or human induced changes in the river, such as the construction of a run-of-the-river power plant cannot only affect its direct vicinity, but also large portions of the surroundings. This is a further important factor besides other human induced changes, such as change in land use (surface sealing, afforestation, deforestation etc.) and pumping activities as for example mentioned by Stoll et al. (2011). In small, and heavily human impacted systems, such as in the Mur valley

described herein, those human induced changes can be among the most important influences, rendering the concept of "natural conditions" almost impossible in shallows wells. Short-term disruptions on the other hand (as demonstrated by well MFd in the Murdurchbruchstal in 2003), do not affect the long term correlations.

## 5   Future work

The correlation matrix approach shown herein could be applied to other regions, since it offers a quick first step to visualize

correlations and thus relations between the different bodies of water found in a region. As we have shown, we see considerable differences between our subregions, even though their aquifer properties are similar (see Section 2.1). However, while we do have a wealth of data available, the aquifer properties are of a rather coarse resolution and thus missing information about possible inhomogeneities. It would be interesting to see whether the differences and similarities identified in our subregions

also hold for different areas in a similar setting (alpine basin, narrow valley, shallow foreland aquifer) and how different settings differ in their relations. Specifically, it would be beneficial to identify a region where more aquifer properties are known in a finer resolution. Also, the apparent differences between unconfined and (semi)confined aquifer bodies warrants further investigation.

Future applications of correlation matrices would also benefit from the inclusion of other phenomena. With the SPEI (Vicente-Serrano et al., 2010) one could already add evapotranspiration to this visualization, which could add valuable insights for many regions. In locations similar to ours, snow plays an important role, as discussed herein in Section 4.3. However, due to its intermittent nature, standardizing snowfall or snow heights is not possible with the approaches used herein. Since the development of new indices was not the focus of this paper, we opted to discuss snow outside of the matrix visualization, but an SSI (Standardized Snow Index) could be a beneficial addition to the hydrologists toolbox. Other possible additions or new indices could be for atmospheric phenomena, such as for example blocking (which is related to cold spells in our region, see Brunner et al. (2017)). Also possible connections between different (sub)regions and the role of rivers as connector of far away regions does warrant further investigation.

The finding that river stages exhibit the highest correlation with groundwater levels in some subregions also warrants further investigation into the causations and mechanisms behind this correlation. A possible start to disentangle the different influences could be using methods such as the Karhunen-Loéve transform, as for example used by Longuevergne et al. (2007) in the Rhine valley aquifer.

Regarding the differences between flood and drought as well as snow rich and snow poor time periods and more generally regarding changes over time (non-stationarity) caused by climate change or more direct human impacts such as hydraulic engineering measures, modelling approaches such as that employed by Peters et al. (2005) or a groundwater model fed with precipitation time series from a local climate model could be used to further assess the feasibility of the matrix approach to detect trends over time.

## 6   Conclusions

Three subregions of the Austrian Mur catchment were analyzed. Long-term time series (1975/1980 - 2010) of 75 groundwater monitoring wells, 9 river gauging stations and 3 regional average precipitation time series have been standardized and correlated in order to gain insight into the controlling factors for groundwater in alluvial aquifers, the effects of extreme events, the impacts of human activities and the development over time. It was shown that the correlation matrix approach enables a quick visualization and comparison of different locations and time spans and that standardized indices, such as the SPI, the SGI and the SRSI (SGI applied to river levels), allow for a thorough comparison of groundwater wells, rivers and precipitation.

With the help of these tools, it was shown that subregions in a catchment can show very different behavior, stemming from their different climatic and geologic conditions as well as human impacts. In general, in small subregions and shallow alluvial aquifers as shown here, the river is always an important driver in the system. As a consequence, (human) impacts on the river (e.g. construction of a run-of-the-river power plant) propagate into the aquifer system. When assessing shallow groundwater

basins in a densely populated area, human impacts must be taken into account. Without this context, many phenomena observed in the system can easily be misinterpreted.

The correlation of standardized groundwater levels with standardized precipitation is more significant in the foreland than in the upstream, Alpine part of the catchment. This corresponds to a tendency towards more shallow water tables in the foreland, and the existence of a second, deeper aquifer in the upstream basin. The shallow wells show time series that are highest correlated with the SPI6, whereas the deep wells show the highest correlation with the SPI12. This highest precipitation - groundwater correlation of the deep wells is still considerably lower than the highest precipitation - groundwater correlation of the shallow wells. Besides being only lowly correlated with precipitation, the deep wells also appear to be unaffected by river stage fluctuations.

Extreme events, exemplified by the 2003 drought,the 2009 floods, the 1985/86 snow rich winter and the 1989/90 snow poor winter, significantly impact the correlations between the standardized time series, but differ in their effects. Drought and snow rich conditions show a tendency towards higher correlations and thus uniform behavior of precipitation, surface water and groundwater, whereas flood and snow poor conditions result in lower correlations and thus irregular behavior. A possible explanation for this observations is the fact that the unconfined aquifers in our subregions have response times of at least one month; therefore short term events, such as floods, will not affect the whole aquifer, whereas events of long duration, such as a drought will propagate through the whole subregion, which will be reflected in the mentioned high correlations. In contrast, the aquifer represented by the deep wells in the Aichfeld subregion are likely confined or semi-confined, which results in much lower response times explaining the consistently high correlations amonng those wells even under flood conditions.

When assessing the development over time, the most recent time period from 1999 to 2010 shows significant changes and a trend towards higher correlations. This corresponds to an increase of the number of events in precipitation with index values of SPI6 below -2 in all subregions and in the groundwater of the foreland subregion, suggesting the increased number of drought events as a possible cause of the observed trend towards higher correlations. The investigated Alpine aquifers, however, exhibit a contrasting behavior with the highest number of negative events in the time before 1986. This suggests that the groundwater levels within these subregions are more strongly influenced by direct human impacts, e.g. on the river, than by changes in precipitation. Thus, direct human impacts must not be ignored when assessing climate change impacts on alluvial aquifers situated in populated valleys. Accounting for human impacts within such assessments remains a challenging task that requires further investigation into the nature of the various impacts and the mechanisms of their propagation through the hydrological system. Further work could address different types of aquifers, including larger aquifer bodies or aquifers in different climate zones.

**Appendix A:  Correlation matrix**

The correlation matrices used in Figures 2 to 5 and Figure 10 in this Appendix are similar to the matrices applied in Stoll et al. (2011) and Loon and Laaha (2015). Each colored rectangle in a matrix refers to a single, color coded Pearson correlation coefficient for two standardized time series.

This approach enables us to split time series into e.g. 12 year periods (as done in Figure 5), pick single years (as done in Figure 3) or to pic arbitrary periods (as done in Figure 4 with a 12 month period spanning from November to October) and to calculate a single Pearson correlation coefficient for those parts of the time series, making it possible to show a development over time (Figure 5), or to compare certain years or periods(Figures 3 and 4). In order to plot all possible combinations of SGI, SPI and SRSI, the matrices have a mirror symmetry, shown by the clearly visible diagonal, which is a representation of the complete correlation of each time series with itself.

The data is sorted from left to right - and top to bottom due to the inherent symmetry - starting with the well that is the furthest away from the Mur in the subregion on its left riverbank with the distance to the river getting smaller until the closest well to the river on its left side is reached, from whereon the distance to the river on its right side starts to increase, ending the groundwater block of the matrix on its right side with the well that is the furthest away from the river on its right riverbank. Following the wells standardized groundwater levels (SGI) the standardized precipitation (SPI) shows the averaging periods of 1, 3, 6, 9 and 12 months (SPI1, SPI3, SPI6, SPI9, SPI12). The final group are the standardized surface water time series (SRSI), showing selected gauging stations of the river Mur in the subregion or tributary streams. In order to enable easier reading of the plot, each group is divided by a blank column (or row).

As discussed above, the top and the left side of Figure 10 are sorted identically. Thus, looking at the first row, its first entry shows the Pearson correlation coefficient for the standardized groundwater time series (SGI) "0" with itself, which is 1. The second entry in the first row is identical to the second entry in the first column, showing the Pearson correlation coefficient for the SGI "0" with SGI "1", which is approximately 0.2. This continues until we reach the 7th entry, which is marked as time series "6" in the example. This is intentionally left blank, to provide some spacing to the group of the entries "7" to "9" which are representations of standardized precipitation with different averaging periods. Here, SPI1 is represented by entry "7", SPI3 by "8" and SPI6 by "9". The 9th entry (time series "10") is again intentionally left blank, to provide some spacing to the single standardized surface water time series (SRSI) in this example. When looking at the second row, it becomes clear that its first entry is the Pearson correlation coefficient for the SGI "0" with SGI "1" (which is the same as the second entry in the first row) and that the second entry in the second row is SGI "1" with itself, which is 1.

Due to this symmetry, the sorting and the spacing between different groups of data, we also get 6 distinctive blocks of correlation coefficients: A - the correlation coefficients for all groundwater wells standardized time series with each other; B - the correlation coefficients for all groundwater wells standardized time series with all standardized precipitation averaging periods; C - the correlation coefficients for all groundwater wells standardized time series with all the standardized surface water gauging stations time series; D - the correlation coefficients for all standardized precipitation averaging periods with each other; E - the correlation coefficients for all standardized precipitation averaging periods with all standardized surface water gauging stations time series and F - the correlation coefficients for all standardized surface water gauging stations with each other.

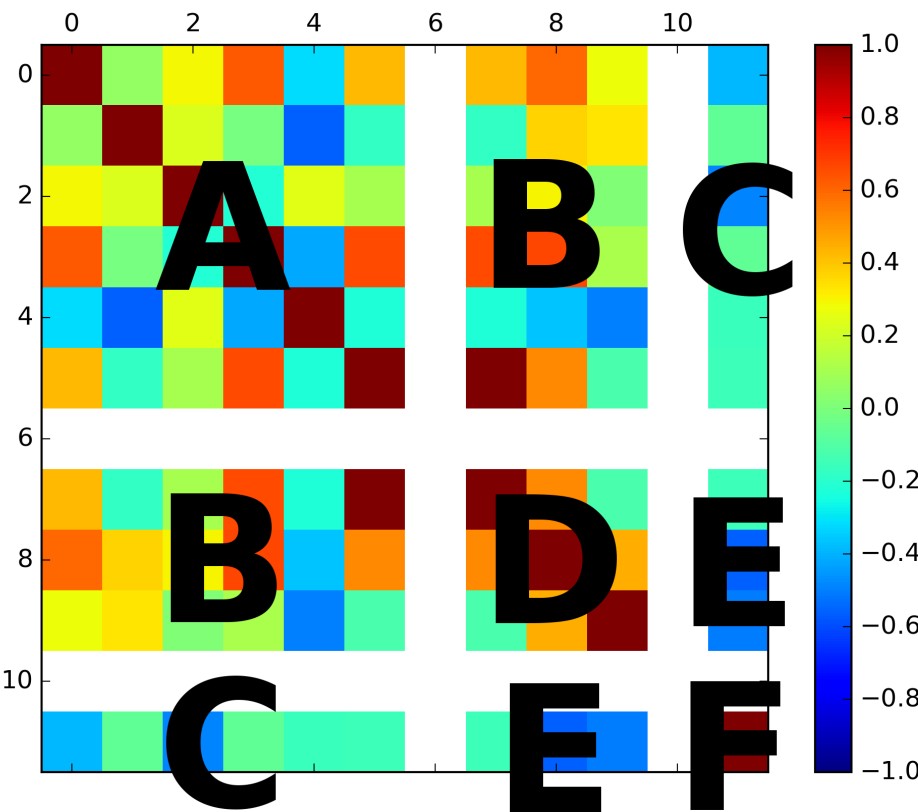

**Figure 10.** Small sample correlation matrix of random data. For a complete explanation, please refer to the text in Appendix A.

## Appendix B: Maps

*Acknowledgements.* This work was funded by the Austrian Science Fund (FWF) under research grant W 1256-G15 (Doctoral Programme Climate Change - Uncertainties, Thresholds and Coping Strategies)

We thank J.P. Bloomfield and B.P. Marchant of the British Geological Survey for their explanations regarding the SGI, M. Switanek of the Wegener Center for climate and global change for his support regarding the SPI and the Department A14 "Wasserwirtschaft, Ressourcen und Nachhaltigkeit" of the Styrian government - most notably B. Stromberger and M. Ferstl - for information regarding the local aquifer systems.

Background maps for Figure 1 and Appendix B: ESRI World shaded relief

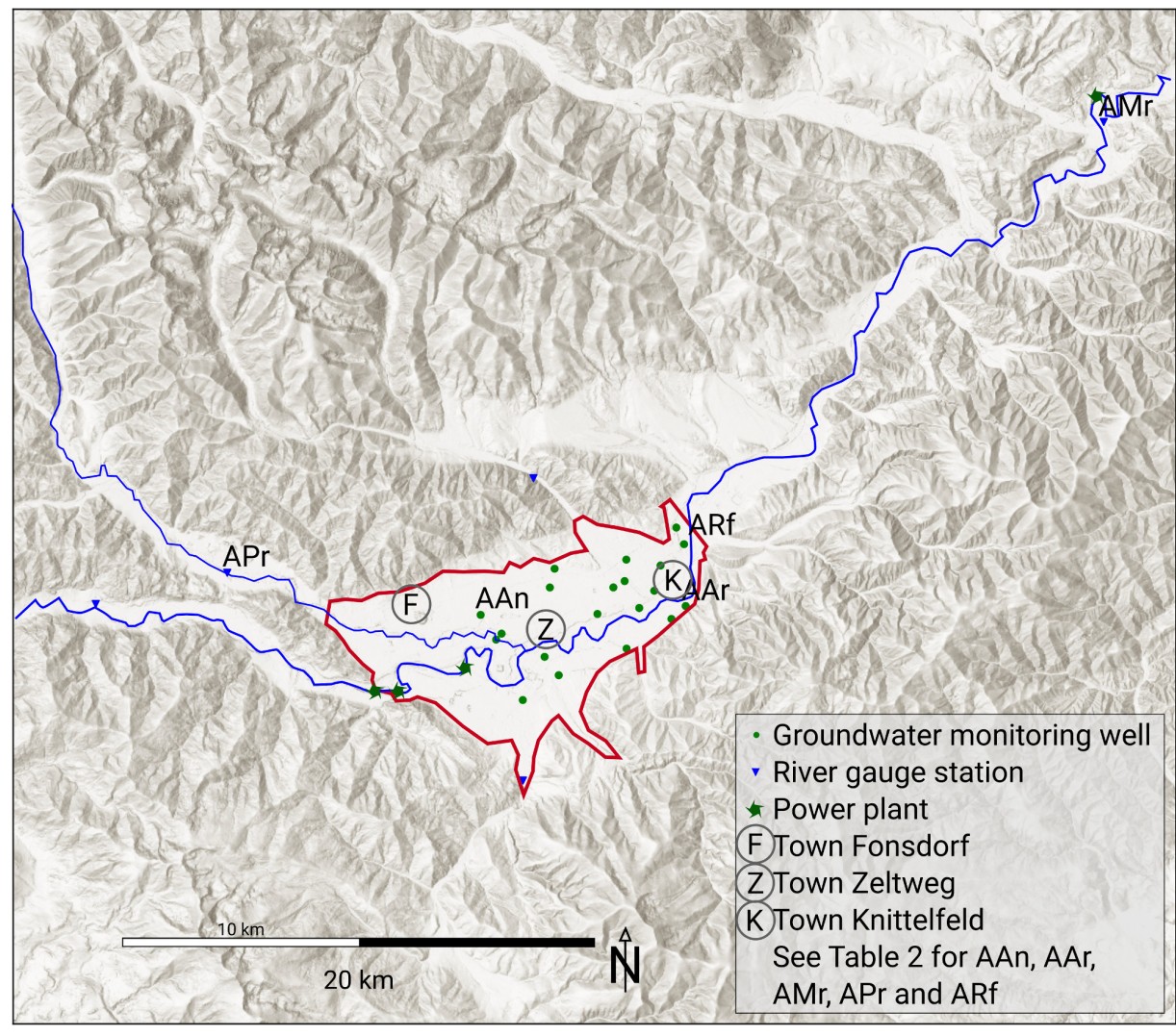

**Figure 11.** Detailed map for the Aichfeld subregion

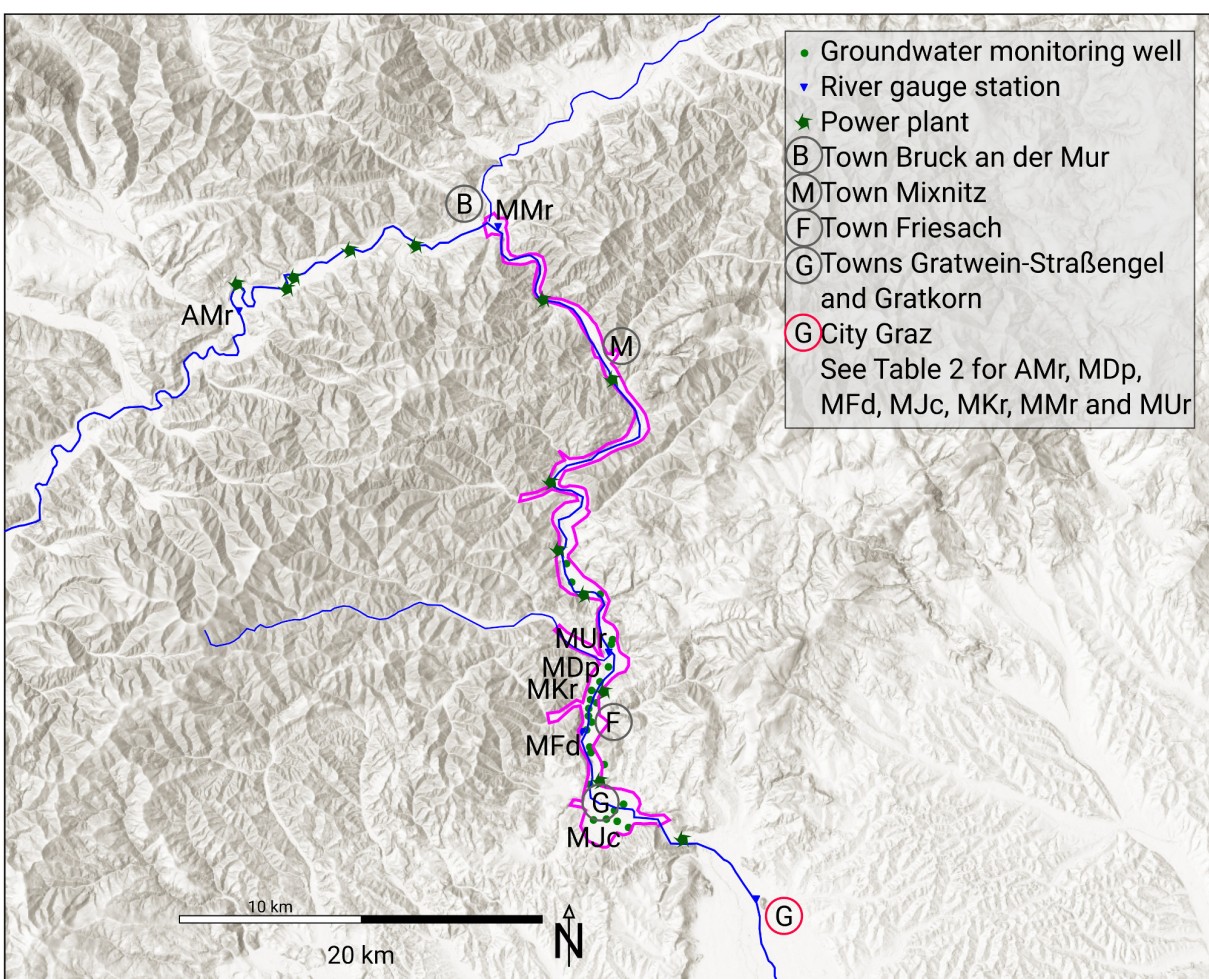

**Figure 12.** Detailed map for the Murdurchbruchstal subregion

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

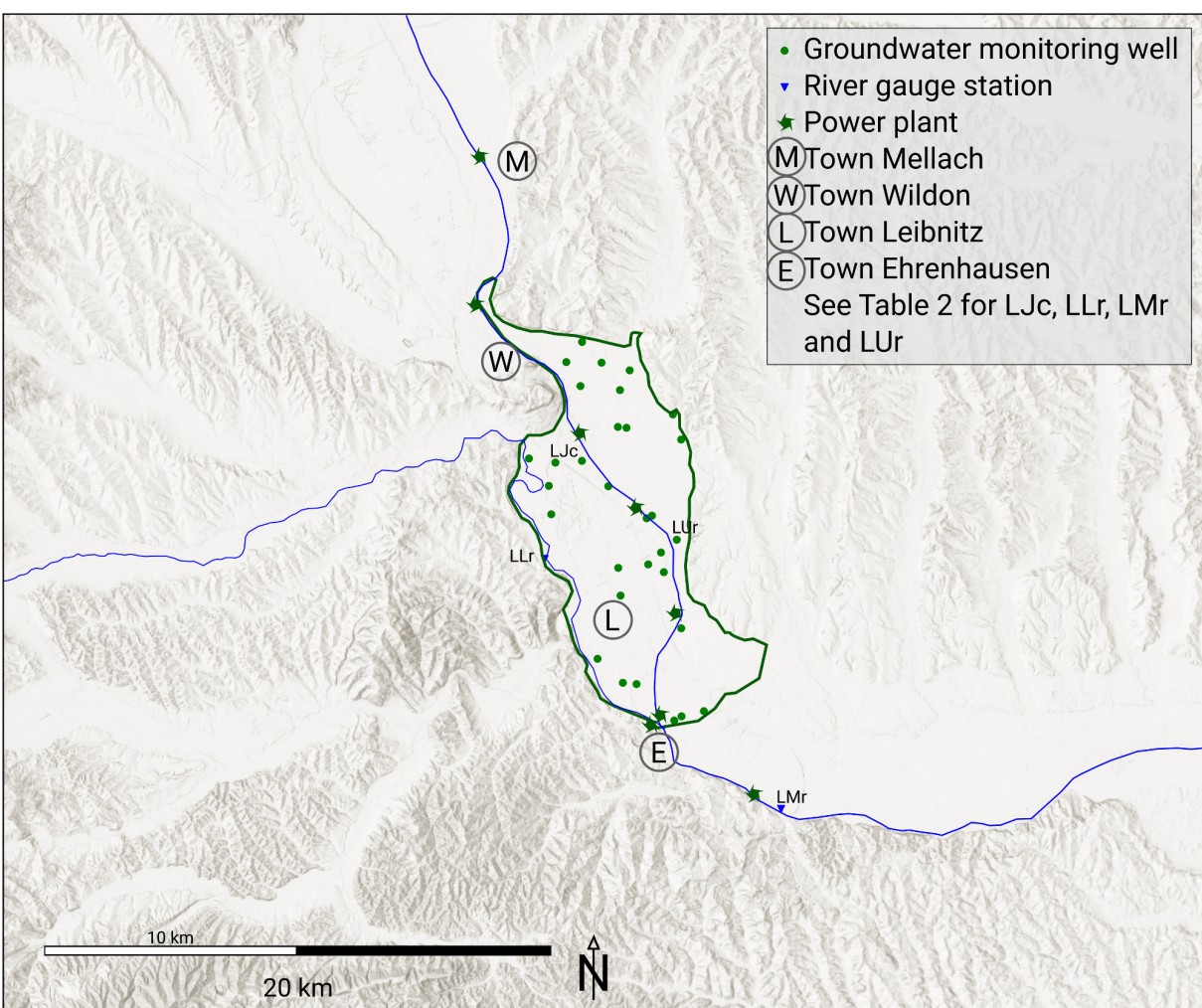

**Figure 13.** Detailed map for the Leibnitzer Feld subregion

Barker, L. J., Hannaford, J., Chiverton, A., and Svensson, C.: From meteorological to hydrological drought using standardised indicators, Hydrology and Earth System Sciences, 20, 2483–2505, doi:10.5194/hess-20-2483-2016, http://www.hydrol-earth-syst-sci.net/20/2483/2016/, 2016.

Benischke, R., Dalla-Via, A., Dobesch, H., Fabiani, E., Fank, J., Fuchs, K., Harum, T., Leditzky, H., Reinsdorff, S., Saccon, P., Schmid, C., Schön, J., Yehdegho, B., and Zojer, H.: Wasserversorgungsplan Steiermark, Ein Leitfaden für die öffentliche Wasserversorgung, vol. 83 of *Berichte der wasserwirtschaftlichen Planung*, http://www.wasserwirtschaft.steiermark.at/cms/dokumente/11913323_102332494/e35d9fbe/83.pdf, 2002.

Beniston, M. and Diaz, H. F.: The 2003 heat wave as an example of summers in a greenhouse climate? Observations and climate model simulations for Basel, Switzerland , Global and Planetary Change, 44, 73–81, doi:10.1016/j.gloplacha.2004.06.006, http://www.sciencedirect.com/science/article/pii/S0921818104000980, extreme climatic events, 2004.

Blain, G. C. and Meschiatti, M. C.: Inadequacy of the gamma distribution to calculate the Standardized Precipitation Index, Revista Brasileira de Engenharia Agrícola e Ambiental, 19, 1129–1135, doi:http://dx.doi.org/10.1590/1807-1929/agriambi.v19n12p1129-1135, 2015.

Bloomfield, J. P. and Marchant, B. P.: Analysis of groundwater drought building on the standardised precipitation index approach, Hydrology and Earth System Sciences, 17, 4769–4787, doi:10.5194/hess-17-4769-2013, http://www.hydrol-earth-syst-sci.net/17/4769/2013/, 2013.

BMLFUW, Abteilung VII 3 - Wasserhaushalt (HZB) im Bundesministerium für Land- und Forstwirtschaft, Umwelt und Wasserwirtschaft: Hydrographisches Jahrbuch von Österreich 2003, vol. 111, lebensministerium.at, https://www.bmlfuw.gv.at/dam/jcr:18fd2126-60a6-4a5e-813a-5f8f9c647f3e/JB2009.pdf, 2006.

BMLFUW, Abteilung VII 3 - Wasserhaushalt (HZB) im Bundesministerium für Land- und Forstwirtschaft, Umwelt und Wasserwirtschaft: Hydrographisches Jahrbuch von Österreich 2009, vol. 117, lebensministerium.at, https://www.bmlfuw.gv.at/dam/jcr:
18fd2126-60a6-4a5e-813a-5f8f9c647f3e/JB2009.pdf, 2011.

BMLFUW, Bundesministerium für Land- und Forstwirtschaft, Umwelt und Wasserwirtschaft, Abteilung IV/4 - Wasserhaushalt (HZB): eHYD HYDROGRAFISCHE DATEN IM INTERNET, https://www.bmlfuw.gv.at/dam/jcr:6bce67b5-52e7-4ab1-8388-27772de65b72/20150421%20eHyd-allgemeine%20Beschreibung_%5BMLO%5D.pdf, accessed January 18, 2017, 2015.

BMLFUW, Bundesministerium für Land- und Forstwirtschaft, Umwelt und Wasserwirtschaft, Abteilung IV/4 - Wasserhaushalt (HZB):
eHYD - Der Zugang zu hydrografischen Daten Österreichs, http://ehyd.gv.at/, accessed July 29, 2016, 2016a.

BMLFUW, Bundesministerium für Land- und Forstwirtschaft, Umwelt und Wasserwirtschaft, Abteilung IV/4 - Wasserhaushalt (HZB): HyDaMS - das Hydrografische Datenmanagement System, https://www.bmlfuw.gv.at/wasser/wasser-oesterreich/wasserkreislauf/hydrographische_daten/HyDaMS.html, accessed January 18, 2017, 2016b.

Brunner, L., Hegerl, G. C., and Steiner, A. K.: Connecting Atmospheric Blocking to European Temperature Extremes in Spring, Journal of
Climate, 30, 585–594, doi:10.1175/JCLI-D-16-0518.1, http://dx.doi.org/10.1175/JCLI-D-16-0518.1, 2017.

Doble, R., Crosbie, R., Peeters, L., Joehnk, K., and Ticehurst, C.: Modelling overbank flood recharge at a continental scale, Hydrology and Earth System Sciences, 18, 1273–1288, doi:10.5194/hess-18-1273-2014, http://www.hydrol-earth-syst-sci.net/18/1273/2014/, 2014.

Doble, R. C., Crosbie, R. S., Smerdon, B. D., Peeters, L., and Cook, F. J.: Groundwater recharge from overbank floods, Water Resources Research, 48, n/a–n/a, doi:10.1029/2011WR011441, http://dx.doi.org/10.1029/2011WR011441, w09522, 2012.

Downing, R., Oakes, D., Wilkinson, W., and Wright, C.: Regional development of groundwater resources in combination with surface water, Journal of Hydrology, 22, 155 – 177, doi:http://dx.doi.org/10.1016/0022-1694(74)90102-4, http://www.sciencedirect.com/science/article/pii/0022169474901024, 1974.

Eltahir, E. A. B. and Yeh, P. J.-F.: On the asymmetric response of aquifer water level to floods and droughts in Illinois, Water Resources Research, 35, 1199–1217, doi:10.1029/1998WR900071, http://dx.doi.org/10.1029/1998WR900071, 1999.

Fabiani, E.: Bodenbedeckung und Terrassen des Murtales zwischen Wildon und der Staatsgrenze, vol. 20 of *Berichte der Wasserwirtschaftlichen Rahmenplanung*, Amt der Steiermärkischen Landesregierung - Landesbaudirektion Wasserbau, http://www.wasserwirtschaft.steiermark.at/cms/dokumente/11913323_102332494/91105bbf/20.pdf, 1971.

Fank, J., Jawecki, A., Nachtnebel, H.-P., and Zojer, H.: Hydrogeologie und Grundwassermodell des Leibnitzer Feldes, vol. 74/1 of *Berichte der Wasserwirtschaftlichen Planung*, Amt der Steiermärkischen Landesregierung - Landesbaudirektion Fachabtteilung IIIa - Wasser-
wirtschaft and Bundesministerium für Land- und Forstwirschaft Wasserwirtschaftskataster, http://www.wasserwirtschaft.steiermark.at/cms/dokumente/11913323_102332494/c9449aec/74-1.pdf, 1993.

García-Herrera, R., Díaz, J., Trigo, R. M., Luterbacher, J., and Fischer, E. M.: A Review of the European Summer Heat Wave of 2003, Critical Reviews in Environmental Science and Technology, 40, 267–306, doi:10.1080/10643380802238137, http://dx.doi.org/10.1080/10643380802238137, 2010.

Gemeinde Knittelfeld: Wasserversorgung Knittelfeld, http://www.knittelfeld.at/leben-in-knittelfeld/umwelt-natur/wasser/, accessed July 29, 2016, 2016.

Godina, R.: Überblick über Daten und Datenarchive im Hydrographischen Dienst für Österreich, in: Niederschlag-Abfluss Modellierung - Simulation und Prognose, edited by Gutknecht, D. and Blöschl, G., vol. 164 of *Wiener Mitteilungen*, pp. 119 – 128, Gutknecht, Dieter, http://www.hydro.tuwien.ac.at/forschung/publikationen/deutschsprachige-publikationen/wiener-mitteilungen-band-164.html, iSBN 3-85234-055-1, 2000.

Guttman, N. B.: Accepting the Standardized Precipitation Index: A Calculation Algorithm, JAWRA Journal of the American Water Resources Association, 35, 311–322, doi:10.1111/j.1752-1688.1999.tb03592.x, http://dx.doi.org/10.1111/j.1752-1688.1999.tb03592.x, 1999.

Hornich, R.: Hochwasser und Hangrutschungen, Wasserland Steiermark, 2, 17–22, http://www.wasserland.steiermark.at/cms/dokumente/10046052_1356921/9f623bc3/wasserland_2_09_web.pdf, 2009.

Koutsoyiannis, D.: HESS Opinions" A random walk on water", Hydrology and Earth System Sciences, 14, 585–601, 2010.

Koutsoyiannis, D.: Hurst-Kolmogorov Dynamics and Uncertainty, JAWRA Journal of the American Water Resources Association, 47, 481–495, 2011.

Kromp-Kolb, H., Nakicenovic, N., Seidl, R., Steininger, K., Ahrens, B., Auer, I., Baumgarten, A., Bednar-Friedl, B., Eitzinger, J., Foelsche, U., Formayer, H., Geitner, C., Glade, T., Gobiet, A., Grabherr, G., Haas, R., Haberl, H., Haimberger, L., Hitzenberger, R., König, M., Köppl, A., Lexer, M., Loibl, W., Molitor, R., Moshammer, H., Nachtnebel, H.-P., Prettenthaler, F., Rabitsch, W., Radunsky, K., Schneider, L., Schnitzer, H., Schöner, W., Schulz, N., Seibert, P., Stagl, S., Steiger, Stötter, H., Streicher, W., and Winiwarter, W.: Synthesis, Tech. rep., Austrian Panel on Climate Change (APCC), 2014.

Kumar, R., Musuuza, J. L., Van Loon, A. F., Teuling, A. J., Barthel, R., Ten Broek, J., Mai, J., Samaniego, L., and Attinger, S.: Multiscale evaluation of the Standardized Precipitation Index as a groundwater drought indicator, Hydrology and Earth System Sciences, 20, 1117–1131, doi:10.5194/hess-20-1117-2016, http://www.hydrol-earth-syst-sci.net/20/1117/2016/, 2016.

Longuevergne, L., Florsch, N., and Elsass, P.: Extracting coherent regional information from local measurements with Karhunen-Loéve transform: Case study of an alluvial aquifer (Rhine valley, France and Germany), Water Resources Research, 43, n/a–n/a, doi:10.1029/2006WR005000, http://dx.doi.org/10.1029/2006WR005000, w04430, 2007.

Loon, A. V. and Laaha, G.: Hydrological drought severity explained by climate and catchment characteristics , Journal of Hydrology, 526, 3–14, doi:10.1016/j.jhydrol.2014.10.059, http://www.sciencedirect.com/science/article/pii/S0022169414008543, drought processes, modeling, and mitigation, 2015.

Lorenzo-Lacruz, J., Morán-Tejeda, E., Vicente-Serrano, S. M., and López-Moreno, J. I.: Streamflow droughts in the Iberian Peninsula between 1945 and 2005: spatial and temporal patterns, Hydrology and Earth System Sciences, 17, 119–134, doi:10.5194/hess-17-119-2013, http://www.hydrol-earth-syst-sci.net/17/119/2013/, 2013.

McKee, T. B., Doesken, N. J., and Kleist, J.: The relationship of drought frequency and duration to time scales, in: Proceedings of the 8th Conference on Applied Climatology, 17-22, pp. 179–183, American Meteorological Society Boston, MA, USA, 1993.

Milly, P. C. D., Betancourt, J., Falkenmark, M., Hirsch, R. M., Kundzewicz, Z. W., Lettenmaier, D. P., and Stouffer, R. J.: Stationarity Is Dead: Whither Water Management?, Science, 319, 573–574, doi:10.1126/science.1151915, http://science.sciencemag.org/content/319/5863/573, 2008.

Müller, G.: Datenprüfung und Verfügbarkeit beim Hydrografischen Dienst in Österreich, in: Methoden der hydrologischen Regionalisierung, edited by Blöschl, G., Godina, R., and Merz, R., vol. 197 of *Wiener Mitteilungen*, pp. 55 – 70, Gutknecht, Dieter, http://www.hydro.tuwien.ac.at/forschung/publikationen/deutschsprachige-publikationen/wiener-mitteilungen-band-197.html, iSBN 3-85234-088-8, 2006.

Nobilis, F. and Godina, R.: Extreme Trockenheit in Österreich, Österreichische Wasser-und Abfallwirtschaft, 58, 51–58, http://link.springer.com/article/10.1007/BF03165684, 2006.

ÖVGW, Österreichische Vereinigung für das Gas- und Wasserfach.: Wasserwerk Graz, Geschichte, http://www.wasserwerk.at/home/wasserwerke/graz/geschichte, accessed July 29, 2016, 2016.

Peters, E., van Lanen, H., Torfs, P., and Bier, G.: Drought in groundwater—drought distribution and performance indicators, Journal of Hydrology, 306, 302 – 317, doi:http://dx.doi.org/10.1016/j.jhydrol.2004.09.014, http://www.sciencedirect.com/science/article/pii/S0022169404004330, 2005.

Ruch, C., Reszler, C., and Schatzl, R.: Operational flood forecasts for the Mur and Enns catchment in Austria - experiences from the June 2009 double flood event, in: EGU General Assembly Conference Abstracts, vol. 12 of *EGU General Assembly Conference Abstracts*, p. 3992, http://meetingorganizer.copernicus.org/EGU2010/EGU2010-3992.pdf, 2010.

Schatzl, R.: Bericht des Hydrografischen Dienstes, Wasserland Steiermark, 2, 13–16, http://www.wasserland.steiermark.at/cms/dokumente/10046052_1356921/9f623bc3/wasserland_2_09_web.pdf, 2009.

Scheucher, P.: Das Ende des Bergbaues in Fohnsdorf - Eine Region im Wandel, Master's thesis, University of Graz, 2004.

Schubert, J.: Hydraulic aspects of riverbank filtration—field studies , Journal of Hydrology, 266, 145–161, doi:10.1016/S0022-1694(02)00159-2, http://www.sciencedirect.com/science/article/pii/S0022169402001592, attenuation of Groundwater Pollution by Bank Filtration, 2002.

Seneviratne, S. I., Luthi, D., Litschi, M., and Schär, C.: Land-atmosphere coupling and climate change in Europe, Nature, 443, 205–209, doi:10.1038/nature05095, http://www.nature.com/nature/journal/v443/n7108/suppinfo/nature05095_S1.html, 10.1038/nature05095, 2006.

Sophocleous, M. A.: Stream-floodwave propagation through the Great Bend alluvial aquifer, Kansas: Field measurements and numerical simulations, Journal of Hydrology, 124, 207–228, doi:10.1016/0022-1694(91)90015-A, http://www.sciencedirect.com/science/article/pii/002216949190015A, 1991.

Stadlbauer, H. and Lorbeer, G.: Grundwasseruntersuchungen in Aichfeld und Pölstal 1992 - 1999, Tech. rep., Amt der Steiermärkischen Landesregierung Fachabteilungsgruppe Landesbaudirektion, Fachabteilung 1a Referat Gewässeraufsicht, http://www.umwelt.steiermark.at/cms/dokumente/10883102_38906880/02a809c7/Aichfeld1992.pdf, 2000.

Stoll, S., Hendricks Franssen, H. J., Barthel, R., and Kinzelbach, W.: What can we learn from long-term groundwater data to improve climate change impact studies?, Hydrology and Earth System Sciences, 15, 3861–3875, doi:10.5194/hess-15-3861-2011, http://www.hydrol-earth-syst-sci.net/15/3861/2011/, 2011.

Stromberger, B., Schatzl, R., and Greiner, D.: Hydrologische Übersicht für das erste Halbjahr 2009, Wasserland Steiermark, 2, 7–11, http://www.wasserland.steiermark.at/cms/dokumente/10046052_1356921/9f623bc3/wasserland_2_09_web.pdf, 2009.

Svoboda, M., Hayes, M., and Wood, D.: Standardized precipitation index user guide, World Meteorological Organization Geneva, Switzerland, 2012.

van der Schrier, G., Efthymiadis, D., Briffa, K. R., and Jones, P. D.: European Alpine moisture variability for 1800–2003, International Journal of Climatology, 27, 415–427, doi:10.1002/joc.1411, http://dx.doi.org/10.1002/joc.1411, 2007.

Vekerdy, Z. and Meijerink, A.: Statistical and analytical study of the propagation of flood-induced groundwater rise in an alluvial aquifer, Journal of Hydrology, 205, 112–125, doi:10.1016/S0022-1694(97)00148-0, http://www.sciencedirect.com/science/article/pii/S0022169497001480, 1998.

VERBUND AG: Kraftwerk Friesach, https://www.verbund.com/de-at/ueber-verbund/kraftwerke/unsere-kraftwerke/friesach, accessed January 18, 2017, 2016a.

VERBUND AG: Kraftwerk Peggau, https://www.verbund.com/de-at/ueber-verbund/kraftwerke/unsere-kraftwerke/peggau, accessed January 18, 2017, 2016b.

Vicente-Serrano, S. M., Beguería, S., and López-Moreno, J. I.: A multiscalar drought index sensitive to global warming: the standardized precipitation evapotranspiration index, Journal of climate, 23, 1696–1718, 2010.

Vicente-Serrano, S. M., López-Moreno, J. I., Beguería, S., Lorenzo-Lacruz, J., Azorin-Molina, C., and Morán-Tejeda, E.: Accurate Computation of a Streamflow Drought Index, Journal of Hydrologic Engineering, 17, 318–332, doi:10.1061/(ASCE)HE.1943-5584.0000433, http://dx.doi.org/10.1061/(ASCE)HE.1943-5584.0000433, 2012.

Wagner, T., Fritz, H., Stüwe, K., Nestroy, O., Rodnight, H., Hellstrom, J., and Benischke, R.: Correlations of cave levels, stream terraces and planation surfaces along the River Mur—Timing of landscape evolution along the eastern margin of the Alps , Geomorphology, 134, 62–78, doi:10.1016/j.geomorph.2011.04.024, http://www.sciencedirect.com/science/article/pii/S0169555X11001930, geomorphology and Natural Hazards in Karst Areas, 2011.

Weider, K. and Boutt, D. F.: Heterogeneous water table response to climate revealed by 60 years of ground water data, Geophysical Research Letters, 37, n/a–n/a, doi:10.1029/2010GL045561, http://dx.doi.org/10.1029/2010GL045561, l24405, 2010.

Workman, S. R. and Serrano, S. E.: Recharge to Alluvial Valley Aquifers from Overbank Flow and Excess Infiltration, JAWRA Journal of the American Water Resources Association, 35, 425–432, doi:10.1111/j.1752-1688.1999.tb03600.x, http://dx.doi.org/10.1111/j.1752-1688.1999.tb03600.x, 1999.

Worsch, E.: Geologie und Hydrologie des Aichfeldes, vol. 25 of *Mitteilungen des Museums für Bergbau, Geologie und Technik am Landesmuseum "Joanneum"*, Murban, Karl, 1963.

ZAMG, Zentralanstalt für Meteorologie und Geodynamik: Klimadaten von Österreich 1971 - 2000, http://www.zamg.ac.at/fix/klima/oe71-00/klima2000/klimadaten_oesterreich_1971_frame1.htm, accessed July 29, 2016, 2016.

Zetinigg, H.: Die Messungen der Fließgeschwindigkeiten des Grundwassers im Mur- und Mürztal, vol. 62 of *Berichte der Wasserwirtschaftlichen Rahmenplanung*, Amt der Steiermärkischen Landesregierung - Landesbaudirektion, Referat für wasserwirtschaftliche Rahmenplanung, http://www.wasserwirtschaft.steiermark.at/cms/dokumente/11913323_102332494/cafb832d/62.pdf, 1982.

Zetinigg, H., Puschnik, T., Novak, H., and Weber, F.: Seismische Untersuchungen im Grundwasserfeld Friesach nördlich von Graz, vol. 7 of *Berichte der Wasserwirtschaftlichen Rahmenplanung*, Amt der Steiermärkischen Landesregierung - Landesbaudirektion Wasserbau, http://www.wasserwirtschaft.steiermark.at/cms/dokumente/11913323_102332494/dc93ebf7/07.pdf, 1966.