# Peer review of "Characterizing the spatiotemporal variability of groundwater levels of alluvial aquifers in different settings using drought indices"

_Hydrology and Earth System Sciences, 2016_

## Referee Comment (RC1) · Anonymous Referee #1 · 30 Sep 2016

General remarks:

1. According to the first sentence of the abstract, the paper aims at "improving the understanding how aquifers in different alluvial settings respond to extreme events in a changing environment". This is a highly relevant and interesting topic. The authors performed a comprehensive correlation analysis of time series of SPI, SGI and SRSI for numerous sites in three different subregions in the Mur catchment. In fact a thorough correlation analysis can be a powerful first step. This study, however, is limited to the description of the correlation matrices and speculation about the reasons of the identified correlation patterns rather than providing a sound test of hypotheses or clearly identifying single processes (see below). The authors often emphasize that observed

similarities might reflect the interplay of various effects. That has been known for long. However, rather than speculating about different effects there is urgent need for disentangling these different effects. Unfortunately, this paper falls short of the mark in this regard.

2. Time series of the SPI were used to assess the role of climatic drivers for groundwater level and river stage dynamics. I would assume that snow melt would play a major role for river discharge and groundwater dynamics in this alpine catchment which is not accounted for, especially for the upstream parts (cf. p. 21, l. 7-8 and p. 24, l. 12-13). Correspondingly, the observed long-term increase of correlation between SPI and SGI or SRSI (Fig. 4) could indicate a long-term tendency of decreasing snow cover. In addition, unlike the SPEI (Vicente-Serrano et al. 2010, Journal of Climatology), SPI does not account for evapotranspiration which is assumed to play an increasing role with regard to drought risk due to climate change.

3. P. 2, l. 6-7: It is stated that "the river is generally found to be a dominant factor, frequently affecting not only the wells closest to the river, but also more distant parts of the alluvial aquifer." Correlation or coincidence does not necessarily imply a causal relationship. In addition, the term "dominant" implies that river stage dynamics would explain a large fraction of the variance in a statistical sense. But that has actually not been quantified in the study. Approaches like that followed by Longuevergne et al. (2007, Water Resources Research) are much more powerful in this regard and would in fact allow to disentangle different effects in a quantitative way.

4. Comparing the correlation matrices for different groups of measurement sites, for the flood year and the drought year, or for different 12-year periods, etc., the stated differences should be tested for significance. The same holds to stated causal relationships, where possible.

Details:

5. P. 4, l. 25: I guess you mean "facies", not "faces".

6. P. 5, l. 20-21: Shouldn't elevation rather decrease from North to South, that is, along the river flow direction?

7. P. 8, l. 16-19: You mean lag times of maximum correlation?

8. P. 8, l. 21-23: The "mirror symmetry" needs not to be mentioned explicitly because it is an inherent feature of correlation matrices.

9. P. 9, caption of figure 2: I suggest to change the first sentence to "Correlation matrices for the three subregions (left panel) and time series of standardized groundwater head, SPI and SRSI (right panel)". The second sentence of the figure caption can be deleted. The labels of the colour scales in the figure are much too small.

10. P. 10, caption of table 2: What do you mean by "precipitation averaging periods"?

11. P. 17, l. 14-20: The authors argue that poor correlation between SPI and SRSI is due to the fact that river stage reflects more patterns of rainfall in upstream parts of the catchment rather than local precipitation. They do not provide any evidence for that. I would assume that snow melt plays a major role in parts of the alpine catchment that is not accounted for by the SPI. In addition, the low-pass filtering of the temporal pattern of rainfall is the strongest in the uppermost soil layer and has a major effect even on the probability distribution of the resulting groundwater and river stage data, thus presumably resulting in poor correlation between SPI and SGI or SRSI.

12. Fig. 5-7: I guess that the respective indices have been determined for each site separately and have then been averaged rather than first averaging the measured values and then determining the index values, is that right? However, even then different periods covered by time series of single sites might introduce some spurious trends. Have you checked for that?

13. P. 23, l. 4-5: What do you mean by "visual similarity"? What aspect of the time series are you talking about?

Linguistic details: Often some rather sloppy and imprecise formulations are used which

is not only annoying but sometimes even confusing or misleading.

14. The plural form of "correlation", "river level", "time" etc. is not adequate. Instead, the terms "correlation coefficients", "time series of river stage", "time periods" etc. should be used. Using the term "rivers" rather than "river gages" (e.g., p. 12, l. 18-19) is not only misleading but false, because it is partly the same river throughout. Correspondingly (p. 12, l. 23-24), not the wells are different but the time series of groundwater head at different sites, and wells cannot be correlated to each other (p. 11, l. 11). It is not the "matrix visualization" that is different but the matrices themselves that are dissimilar (p. 12, l. 21-22).

15. P. 3, l. 9-10: What do you mean by a "starting data set"?

16. P. 17, l. 35: Please replace "depth" by "thickness".

17. P. 18, l. 8-10: Please replace "decrease" by "lower correlation coefficients". The term "decrease" describes a more or less monotonic change over time which is not adequate when you compare two distinct time periods without considering the transition period in between.

18. P. 22, l. 10: Please replace "culprit" by "reason".

19. P. 24, l. 22-24: What is a "negative event"? Do you mean "periods with negative index values"?
* * *

---

## Referee Comment (RC2) · J. Huntington (Referee) · 10 Dec 2016

The manuscript aims to improve the understanding how aquifers in different alluvial settings respond to extreme events through the use of standardized time series of groundwater levels (SGI), precipitation (SPI), and river stages (SRSI) for three study areas within the river Mur river basin. Using correlation matrices to visualize results, differences and similarities of the study areas are discussed.

General Comments: I am impressed by the richness of the groundwater level data in this region, and that is used in the manuscript. Also, I enjoyed learning about the area. The paper presents a novel way to view water level correlations to each other, and physical drivers via SPI and SRSI, for potentially understanding aquifer responses

to extreme events, human influences, and provides potential to gain insight of aquifer properties.

One of my main concerns is that the correlation matrix figures, which are the primary results figures, are extremely hard to interpret given the current figure symbology, and figure discussion. There should be very clear descriptions given on how to interpret these figures when introduced, and also more during figure interpretation and discussion when detailing results.

I am disappointed that the words hydraulic conductivity, transmissivity, or permeability, are not found in the paper, nor are there discussions on how these aquifer properties (and aquifer storage) influence the results. It is recommended that the authors relate the results to aquifer properties. If property information is no available, then it is suggested that proxies and generalized properties should be used to make the results and conclusions more complete.

Monthly correlations for different time scales of SPI were not considered, which could show additional details on recharge, discharge, and surface and groundwater interactions. Authors should consider the addition of monthly correlations similar to methods presented in references pointed out in specific comments.

Perhaps the addition of a discussion on how a combination of modeling and this approach could be used to calibrate and better understand the human / landuse / pumping impacts surface and groundwater interactions might be a good addition.

Specific Comments

-Many paragraphs are small/one sentence paragraphs. These should be combined together.

-Page 1; line 1. Suggest adding "of" in "To improve the understanding (of) how aquifers.."

-Precipitation measurements collected in valley bottoms, and assumed uniform over

each sub basin. What about spatial and temporal aspects of this assumption? Please mention.

-Page 6; line 24. "While there is some criticism of the gamma distribution (see e.g. Guttman (1999)), it is generally a widely used and recommended mended index (see e.g. Svoboda et al. (2012))."

Good point - there have been quite a few questions raised lately about issues of the SPI's assumed gamma distribution assumption – perhaps cite this new one and discuss – and if these findings perhaps influence yours.

Blain, G. C., & Meschiatti, M. C. (2015). Inadequacy of the gamma distribution to calculate the Standardized Precipitation Index. Revista Brasiliera de Engenharia Agrícola e Ambiental, 19(12), 1129-1135. http://www.scielo.br/scielo.php?pid=S1415-43662015001201129&script=sci_arttext

-Page 3; line 8. "For these three subregions monthly groundwater levels as well as river stages and precipitation are available at a the ehyd.gv.at website (BMLFUW, 2016)."

How are groundwater levels measured – steel tape, e-tapes, pressure transducers? What type of precipitation data (i.e. snow, rain, both), and how is it measurements? Please provide more background and details. Very impressive that all this historical data exists, but more background would be great to get a better picture of how it is done and perhaps other agencies can adopt.

-Page 8; line 10. "For each possible combination of standardized wells (SGI), standardized precipitation (SPI) or standardized river stages (SRSI) a Pearson Correlation coefficient was calculated. In order to facilitate the comparison of standardized groundwater levels, river stages, and precipitation within the individual subregions, the correlations between the indices have been plotted within a matrix, showing all the groundwater monitoring wells, all the river stages and SPI1, 3, 6, 9 and 12 for each subregion, similar to the matrices applied in Stoll et al. (2011) and Loon and Laaha

(2015)."

More details need to be discussed so the reader can get a better understanding of the matrix plots – like "for example, along the x-axis is …. where each cell represents a different …" etc.. Perhaps follow the descriptions by Stoll and Loon and Laaha to provide further explanations.

-Page 8; line 16 – "According to Vekerdy and Meijerink (1998), correlations between daily river stages and groundwater levels in distances similar to those relevant for this paper are mostly below 30 days. Likewise, Bloomfield and Marchant (2013) as well as Kumar et al. (2016) found with few exceptions the highest correlation between SGI and SPI associated with a time lag of zero months. As this is particularly expected in shallow alluvial aquifers, only Pearson Correlation coefficients without a time lag are considered here."

For all wells? Did you explore if optimal time scales were greater than or less than 1 month? This would be a fairly easy thing to explore, especially put into context of well distance from the stream, etc. Presenting correlations without time lags seems odd.

-Figure 2. This is really complex to interpret. Is there any additional labeling / symbology that could help? For example I count 5 boxes under the SPI 1-12, and SPSI has 3 boxes. Can you help the reader interpret the figure better? For example the diagonal – can you tell the reader what this means? What do the "bulls eyes" correlation features represent? The figure caption should be pretty self-descriptive, so this one could be long – or integrated into text.

-Page 10; line 6. "Most wells outside of the core of this region show a similar behavior, resulting in an average Pearson correlation coefficient of all of these wells with each other of 0.59. These wells show a low correlation with the SPI1 and moderate to high correlations with the longer SPI averaging periods, as expected from the previous literature (Bloomfield and Marchant, 2013; Kumar et al., 2016)."

Perhaps add that this is to be expected since these wells are further away?

-Page 11; line 6. The wells from the deeper aquifer also show a clear increase in correlation with an increase in the length of the SPI averaging periods, starting with an average correlation of the wells with the SPI1 of -0.04, reaching a maximum correlation of 0.38 with the SPI12, which is significantly lower than the correlations seen in the shallow wells. The average correlations of the deeper wells with the rivers range from -0.13 with the local Pöls to 0.24 with the downstream Mur.

Which deep well/labels should the reader be looking at exactly in Figure 2 to see this longer / smoother water level signal?

-Page 11; line 11. "The rivers are correlated well with each other, indicating a similar flow regime in the upstream and downstream Mur, as well as in the tributary Pöls, but the correlations with the precipitation are low to moderate, ranging form an average of 0.27 with SPI1 to 0.48 with SPI6."

-Perhaps the low correlation is the assumption of a standardized time scale of 0 months for stream stage?

Can you describe the impact of snow accumulation and melt, and possible lag times in this statement?

See McEvoy et al. (2012) and Abatzoglou et al. (2014) on evaluating precipitation and streamflow indices for different time scales.

McEvoy, D. J., Huntington, J. L., Abatzoglou, J. T., & Edwards, L. M. (2012). An evaluation of multiscalar drought indices in Nevada and Eastern California. ÂăEarth Interactions, Âă16(18), 1-18.

Abatzoglou, J. T., Barbero, R., Wolf, J. W., & Holden, Z. A. (2014). Tracking interannual streamflow variability with drought indices in the US Pacific Northwest. ÂăJournal of Hydrometeorology, Âă15(5), 1900-1912.

-Page 12; line 1. "Surprisingly, some of the wells closest to the Mur on both sides of the river are not very well correlated with each other and are also not among the wells with highest correlations with the rivers."

Is this due to differences depth of well screen intervals / depths? Please explain.

-Page 12; Line 14. "Well MKr is located closest to the river Mur, yet it shows no high correlation with it."

Why is this? What is the well depth of MKr? Can you tell the reader what your idea is about why this is?

-Page 12; line 18. "The rivers are very highly correlated with each other, but only show some minor correlations with the 3 and 6 month SPI with average correlation coefficients of 0.38 and 0.39."

Perhaps because monthly correlations were not considered? See figures 9 and 10 and respective discussions in McEvoy et al. (2012) for some ideas to further describe why correlations were low, or if analyzed a different way, may increase. https://www.researchgate.net/publication/236687241_An_Evaluation_of_Multi-scalar_Drought_Indices_in_Nevada_and_Eastern_California

-Page 16; line 2. "Figure 4 shows the development of the three subregions when split-up into time periods of 12 years."

How so? Please further explain figure 4 to this point.

-Page 17; line 29. "At high water levels, the river feeds the groundwater, thus superpositioning its signal onto the groundwater, whereas the groundwater provides the river baseflow in low water conditions, thus giving the river a groundwater signal at low water levels."

Can you show this point with the standardized time series? It is not clear from the correlation matrix plots. What is the "groundwater signal at low water levels." Is there a

"signal" during this condition?

-Page 18; line 25 – suggest using the term groundwater discharge rather than groundwater runoff.

-Figure 5. What time scale are the SPI values – 0 month? Please specify.

-Figure 5 and 6. Why not use 48 or 60 month time scales for SGI, SPI, and SRSI rather than using a 0 month and then smooth using a 5 year moving average of the indices? Seems a bit odd and misses the point of the use of multi-temporal time scale indices. Please explain.

-Page 22; line 4 – change hydro(geo)ology to hydrogeology. Also, it is not agreed that the "general consensus is that hydrogeology is stationarity." Milly et al., 2008 states that stationarity it is dead – but that in water management stationarity is often assumed. . . this is not the same as a "general consensus in hydrogeology seems to be the assumption of stationarity" as stated - please revise.

-Page 24; line 3. "It was shown that the correlation matrix approach enables a quick visualization and comparison of different locations and time spans and that standardized indices, such as the SPI, the SGI and the SRSI (SGI applied to river levels), allow for a thorough comparison of groundwater wells, rivers and precipitation."

More thorough labeling and discussion on how to interpret the matrix plots when they are introduced, and during results discussion is needed for these plots to "enable a quick visualization and comparison of different locations and time spans" – please provide more details to help the reader digest these plots.

---

## Author Response (AR1)

We would like to thank the Editor for pointing out two interesting areas to focus on. Regarding the first one, we added Section 5, Future Work, which – besides some general suggestions – points out specific issues raised in the review (addition of indices such as the SPEI, further work on snow) and issues identified thanks to the review (need for more aquifer properties and testing it in regions with differing aquifer properties). The changes made in response to the suggestions by the Editor and the reviewers have also helped providing a clearer description of the new knowledge provided by our paper, and this is further supported by additional modifications in the abstract and the conclusion section.

Regarding the second comment by the Editor and the related Referee comment, we have added the available information about aquifer properties to Section 2.1. In general, our subregions are unconfined and of a similar hydraulic conductivity, but given their sedimentation history, certain irregularities are to be assumed, but can not be pinpointed to a specific time series. This additional data enabled us to add the concept of aquifer response time to the discussion given in Sections 4.3 and 4.4, which offers an explanation for the changes in correlations observed under drought, flood, snow rich and poor conditions. The small subset of the deep wells in the Aichfeld subregion is likely confined or semi confined, so the resulting expected increase in diffusivity and thus reduction in response times offers an explanation for the consistent high correlations of those wells SGI time series, especially so under flood conditions (see Section 4.3). This is now also highlighted in the conclusions.

We have updated our manuscript, following the comments of the reviewers and our replies to them. In our manuscript, the changes are highlighted in red and blue. Further – where applicable – we have added green markers to make it easier to link a change to a specific comment by Referee #1 (C1-...) and Referee #2 (C2-...).

Following below, are the original comments (**C:** *cursive*) and our replies to them (**R:** normal font) and further comments regarding the implemented changes as well as page and line numbers where applicable (**R:** blue font).

**Reply to Anonymous Referee #1,:**

**C:***1. According to the first sentence of the abstract, the paper aims at "improving the understanding how aquifers in different alluvial settings respond to extreme events in a changing environment". This is a highly relevant and interesting topic. The authors performed a comprehensive correlation analysis of time series of SPI, SGI and SRSI for numerous sites in three different subregions in the Mur catchment. In fact a thorough correlation analysis can be a powerful first step. This study, however, is limited to the description of the correlation matrices and speculation about the reasons of the identified correlation patterns rather than providing a sound test of hypotheses or clearly identifying single processes (see below). The authors often emphasize that observed similarities might reflect the interplay of various effects. That has been known for long. However, rather than speculating about different effects there is urgent need for disentangling these different effects. Unfortunately, this paper falls short of the mark in this regard.*

**R:** The aim of our paper is to provide a tool for this important first step by enabling a qualitative view on differences and similarities, instead of discussing all the possible phenomena. The disentangling can be a following step (see also C2 and C6 by reviewer 2).

However, we do agree that there is a need to add some more quantitative information (see also C4 by reviewer 1) and more care in the wording (see C3 by reviewer 1). Thus we will revise parts of the manuscript accordingly.

**R:** We have added quantitative information about the significance of differences and changes of the correlations and about the aquifer properties including a brief discussion on aquifer response times. See the responses below for more details.

**C:**2. Time series of the SPI were used to assess the role of climatic drivers for groundwater level and river stage dynamics. I would assume that snow melt would play a major role for river discharge and groundwater dynamics in this alpine catchment which is not accounted for, especially for the upstream parts (cf. p. 21, l. 7-8 and p. 24, l. 12-13). Correspondingly, the observed long-term increase of correlation between SPI and SGI or SRSI (Fig. 4) could indicate a long-term tendency of decreasing snow cover. In addition, unlike the SPEI (Vicente-Serrano et al. 2010, Journal of Climatology), SPI does not account for evapotranspiration which is assumed to play an increasing role with regard to drought risk due to climate change.

**R:** We do agree that snow could be an important factor when assessing the long term development of the area. Nevertheless, we decided to omit snow cover since it is an intermittent dataset by its nature and thus not fitting for our chosen method of using correlation matrices. It is expected that the SPI values based on averaging periods of several months (e.g. SPI6) are unaffected by intermittent snow cover and melt effects, probably in part explaining their higher correlations with SGI (compared with shorter averaging periods, e.g. SPI1). We intend to look more closely at the snow data of the study areas, and will add this to the discussion of our results and observations.
We also agree that, in general, changes in evapotranspiration potentially affect groundwater levels. Temperature-based approaches, such as that included in the SPEI, can be used for considering these effects in a simplified way, although it is known that other factors (e.g., wind speed, response of vegetation to climate change, etc.) may be important when assessing changes in evapotranspiration (see e.g. "Global review and synthesis of trends in observed terrestrial near-surface wind speeds: Implications for evaporation," McVicar et al., J. Hydrol., 2012 and "Projected increase in continental runoff due to plant responses to increasing carbon dioxide," Betts et al., Nature, 2007). Ignoring other effects, higher temperature would result in higher evapotranspiration and thus potentially less correlation of SPI with SGI. In the given case, however, the Leibnitzer Feld area shows the highest temperature, but also the highest correlations between SPI and SGI. Which suggests that evapotranspiration cannot explain the observed differences between the study areas. We intend to add a brief discussion of this subject to the paper.

**R:** We have added an extensive discussion of snow to our manuscript, mainly regarding the effects of snow rich and snow poor years. Most of this is found in the new Sections 3.3 and 4.3. Since snow data is not feasible to be used with the standardization approach we used, we also suggest that a Standardized Snow Index could be developed (see Section 5).
Whereas the discussion of snow provides important additional insight into the precipitation data and the corresponding index SPI, a more extensive discussion of evapotranspiration and the SPEI is beyond the scope of this paper. We are interested in the propagation of drought from the atmosphere (and river) to the groundwater. The soil is an important intermediate step and closely related to actual evapotranspiration. However, as the data set to our disposal does not provide soil moisture data, any discussion about actual evapotranspiration would remain speculative. As mentioned in our answer above, we agree that exploring the SPEI generally might be interesting, but since this paper is focused on data analysis rather than on the development or assessment of indices (apart from those directly representing the data used herein) we only refer to this as future work (see Section 5).

**C:**3. P. 2, l. 6-7: It is stated that "the river is generally found to be a dominant factor, frequently affecting not only the wells closest to the river, but also more distant parts of the alluvial aquifer." Correlation or coincidence does not necessarily imply a causal relationship. In addition, the term "dominant" implies that river stage dynamics would explain a large fraction of the variance in a statistical sense. But that has actually not been quantified in the study. Approaches like that followed by Longuevergne et al. (2007, Water Resources Research) are much more powerful in this regard and would in fact allow to disentangle different effects in a quantitative way.

**R:** We assume the reviewer is talking about lines 6-7 on page 1.We agree that this sentence can be worded more carefully. We suggest changing it to "river stages exhibit highest correlations with groundwater levels, frequently affecting…"
Applying methods like the KLT described by Longuevergne et al. would be a promising next step, but this is beyond the scope of this paper.

**R:** Changed as indicated in our answer above. See page 1, lines 6-7 and Section 5.

**C:**4. Comparing the correlation matrices for different groups of measurement sites, for the flood year and the drought year, or for different 12-year periods, etc., the stated differences should be tested for significance. The same holds to stated causal relationships, where possible.

**R:** We agree that testing the significances of the differences would be a valuable addition to the manuscript. We will revise it accordingly.

**R:** We have added extensive information about the significances of mentioned changes (t-tests). Further we have tried to standardize our wording and have thus changed, added or removed some words throughout the manuscript.
In order to have the changes and significances at a convenient location, we have added Table 5 for the changes between the three time periods and Tables 3 and 4 for the changes between flood and drought and the newly added snow rich and snow poor years.

**C:**5. P. 4, l. 25: I guess you mean "facies", not "faces".

**R:** In this area, the valley sides are partly made up out of cliffs and rock faces, thus the use of the term "face". To avoid misunderstandings, we would suggest changing the sentence to "From the town of Bruck an der Mur at the beginning, the valley is incised into metamorphic gneisses, amphibolites and shists of the Austroalpine crystalline basement."

**R:** Changed as indicated in our answer above. See page 5, line 6.

**C:**6. P. 5, l. 20-21: Shouldn't elevation rather decrease from North to South, that is, along the river flow direction?

**R:** Thank you for pointing out this error. The terms "northern" and "southern" got erroneously switched in our manuscript and will be changed.

**R:** Changed as indicated in our answer above. See page 6 line 7.

**C:**7. P. 8, l. 16-19: You mean lag times of maximum correlation?

**R:** Yes. We would suggest changing the first sentence to "According to Vekerdy and Meijerink (1998), highest correlations between daily river stages and groundwater levels in distances similar to those relevant for this paper are mostly found for lag times below 30 days"

**R:** We have implemented the change (see page 9, line 18 ff) and added a discussion of lag times and our reasons to use no lag times to this Section 2.3.

**C:**8. P. 8, l. 21-23: The "mirror symmetry" needs not to be mentioned explicitly because it is an inherent feature of correlation matrices.

**R:** While this is the case, it seems that correlations matrices might require some more explanation. We suggest wording our description in the main body of text more concisely, while adding an appendix, with more information about correlation matrices. See also our reply to C3 and C14 by reviewer 2.

**R:** We have shortened the section in question (see Section 2.3), as well as the description of Figure 2 regarding the description of the correlation matrices, but have added an extensive description in the new Appendix A.

**C:**_9. P. 9, caption of figure 2: I suggest to change the first sentence to "Correlation matrices for the three subregions (left panel) and time series of standardized groundwater head, SPI and SRSI (right panel)". The second sentence of the figure caption can be deleted. The labels of the colour scales in the figure are much too small._

**R:** We agree that the labels or the color scales and some other aspects of the figure(s) are too small and will change them accordingly. For the other issues, please see above.

**R:** Changed as indicated in our answer above. See Figure 2.

**C:**_10. P. 10, caption of table 2: What do you mean by "precipitation averaging periods"?_

**R:** By this we mean the averaging periods in the SPI. We will check the manuscript for consistency in the use of "Precipitation" and "SPI".

**R:** We have added this term (as introduced in the McKee (1993) paper) to section 2.2.1 (see page 8, line 16), so that it's use/meaning will be clearer.

**C:**_11. P. 17, l. 14-20: The authors argue that poor correlation between SPI and SRSI is due to the fact that river stage reflects more patterns of rainfall in upstream parts of the catchment rather than local precipitation. They do not provide any evidence for that. I would assume that snow melt plays a major role in parts of the alpine catchment that is not accounted for by the SPI. In addition, the low-pass filtering of the temporal pattern of rainfall is the strongest in the uppermost soil layer and has a major effect even on the probability distribution of the resulting groundwater and river stage data, thus presumably resulting in poor correlation between SPI and SGI or SRSI._

**R:** We will address snow (see also C2 by reviewer 1) and further investigate the connection between the subregions. Evidently, the runoff process and thus the correlation between SPI and SRSI is also influenced by the soil. We agree that the contribution of the various factors cannot be disentangled with the data and approaches used in the paper and therefore will consider rewording this passage.

**R:** We have added some discussion of the connection of the different stretches of river to Section 4.1 (see page 23, line 10 ff.). We found that the differences between the different gauging stations are not significant, which we deem to support our argument. Further, we have added an extensive discussion of snow to our manuscript, mainly regarding the effects of snow rich and snow poor years. Most of this is found in the new Sections 3.3 and 4.3.

**C:**_12. Fig. 5-7: I guess that the respective indices have been determined for each site separately and have then been averaged rather than first averaging the measured values and then determining the index values, is that right? However, even then different periods covered by time series of single sites might introduce some spurious trends. Have you checked for that?_

**R:** The assumption for the calculation of the average values is correct. There is indeed a risk that the beginning and the end of the plots is dominated by a small number of time series, which might introduce some trend that is not valid for the whole region. We suggest that we either mention this possible issue in a more obvious way, or that we redo the plots with the same dataset used in all other (matrix) plots.

**R:** We have redone the plots (now Figures 6 – 8), using the same dataset used for the matrices. Additionally, we have added a Figure 9, showing the development of snow fall and snow heights for the same time period.

**C:**_13. P. 23, l. 4-5: What do you mean by "visual similarity"? What aspect of the time series are you talking about?_

**R:** We are talking about the fact that the time series discussed by Koutsoyiannis (figure 3 in the cited paper) looks similar to our real time series. Of course the appearance of a plotted time series is a result of its values, so we suggest to remove the word "visual" and to add some more information on Koutsoyiannis work.

**R:** We have removed the word "visual" and added some extensive discussion of the cited paper, to better explain our point (see page 31, line 15 ff.). We also suggest further work on this apparent non-stationarity in our new Section 5.

**C:** *Linguistic details: Often some rather sloppy and imprecise formulations are used which is not only annoying but sometimes even confusing or misleading.*

**R:** We have addressed the issues raised in the specific comments above and below, and will check the manuscript for similar formulations.

**R:** We have checked the whole manuscript for imprecise formulations, such as those mentioned in the following comment, and have made numerous changes.

**C:** *14. The plural form of "correlation", "river level", "time" etc. is not adequate. Instead, the terms "correlation coefficients", "time series of river stage", "time periods" etc. should be used. Using the term "rivers" rather than "river gages" (e.g., p. 12, l. 18-19) is not only misleading but false, because it is partly the same river throughout. Correspondingly (p. 12, l. 23-24), not the wells are different but the time series of groundwater head at different sites, and wells cannot be correlated to each other (p. 11, l. 11). It is not the "matrix visualization" that is different but the matrices themselves that are dissimilar (p. 12, l. 21-22).*

**R:** We will address these issues, and search for similar ones.

**R:** We went through the whole manuscript and used concise wording for issues raised above.

**C:** *15. P. 3, l. 9-10: What do you mean by a "starting data set"?*

**R:** The historical development of the data set available at ehyd.gv.at which is tied to the historical development of water supply in Austria. We will rephrase this and add some more information. See also C11 by reviewer 2.

**R:** We have added some further information and sources on the development and management of the ehyd.gv.at system to section 2.1. See page 3, line 11 ff.

**C:** *16. P. 17, l. 35: Please replace "depth" by "thickness".*

**R:** Yes.

**R:** Changed as indicated in our answer above. See page 24, line 2.

**C:** *17. P. 18, l. 8-10: Please replace "decrease" by "lower correlation coefficients". The term "decrease" describes a more or less monotonic change over time which is not adequate when you compare two distinct time periods without considering the transition period in between.*

**R:** Thank you for pointing out this imprecise formulation. We will change the sentence accordingly.

**R:** Changed as indicated in our answer above. See Section 4.2.

**C:** *18. P. 22, l. 10: Please replace "culprit" by "reason".*

**R:** Yes

**R:** Changed as indicated in our answer above. See page 32, line 7.

**C:** *19. P. 24, l. 22-24: What is a "negative event"? Do you mean "periods with negative index values"?*

**R:** Yes, or, to be more specific, "periods with index values below -2 in SPI6" as discussed in section 4.3.
We will reword the sentence to be more precise.

**R:** Changed as indicated in our answer above. See page 34, line 8 ff.

**Reply to J. Huntington (Referee),**:

**C1:** *The manuscript aims to improve the understanding how aquifers in different alluvial settings respond to extreme events through the use of standardized time series of groundwater levels (SGI), precipitation (SPI), and river stages (SRSI) for three study areas within the river Mur river basin. Using correlation matrices to visualize results, differences and similarities of the study areas are discussed.*

**C2:** *General Comments: I am impressed by the richness of the groundwater level data in this region, and that is used in the manuscript. Also, I enjoyed learning about the area. The paper presents a novel way to view water level correlations to each other, and physical drivers via SPI and SRSI, for potentially understanding aquifer responses to extreme events, human influences, and provides potential to gain insight of aquifer properties.*

**C3:** *One of my main concerns is that the correlation matrix figures, which are the primary results figures, are extremely hard to interpret given the current figure symbology, and figure discussion. There should be very clear descriptions given on how to interpret these figures when introduced, and also more during figure interpretation and discussion when detailing results.*

**R:** This statement contrasts C8 by reviewer 1. It seems that correlation matrices are either a well known tool, or totally unknown, which also is in agreement with our experiences when discussing them at various meetings (e.g. EGU and AGU). Thus we suggest providing only a short description in the main text and a more thorough appendix explaining the method in detail, so that both groups are addressed.

**R:** We have shortened the section in question, as well as the description of Figure 2 regarding the description of the correlation matrices, but have added an extensive description in the new Appendix A.

**C4:** *I am disappointed that the words hydraulic conductivity, transmissivity, or permeability, are not found in the paper, nor are there discussions on how these aquifer properties (and aquifer storage) influence the results. It is recommended that the authors relate the results to aquifer properties. If property information is no available, then it is suggested that proxies and generalized properties should be used to make the results and conclusions more complete.*

**R:** We will peruse the publications about the areas and add some information on those and their relation to the observed phenomena. However, most of the literature seems to be old geologic surveys and government reports.

**R:** We have added the results of various reports about the areas to sections 2.1.1 – 2.1.3. Mainly, the available data concerns hydraulic conductivity and (saturated) thickness of the aquifers.
The addition of this also enabled us to calculate aquifer response times, which greatly improves the discussion of flood and drought years (Section 4.2) and the newly added snow rich and snow poor years (Section 4.3).

**C5:** *Monthly correlations for different time scales of SPI were not considered, which could show additional details on recharge, discharge, and surface and groundwater interactions. Authors should consider the addition of monthly correlations similar to methods presented in references pointed out in specific comments.*

**R:** We do consider different averaging periods of SPI; perhaps the reviewer refers to the fact that correlations without lag time are presented? If so, see our response to C13 for this issue.

**C6:** *Perhaps the addition of a discussion on how a combination of modeling and this approach*

*could be used to calibrate and better understand the human / landuse / pumping impacts surface and groundwater interactions might be a good addition.*
*Specific Comments*

**R:** We are currently working on these issues, but that would be out of scope of the current work. We will however add this as a possible further work at the end of the paper.

**R:** We have added a Section 5, discussing future work.

**C7:** *-Many paragraphs are small/one sentence paragraphs. These should be combined together.*

**R:** We will consider merging some of the short paragraphs.

**R:** We have merged some paragraphs.

**C8:**-*Page 1; line 1. Suggest adding "of" in "To improve the understanding (of) how aquifers.."*

**R:** We agree that this addition would improve the readability of the sentence.

**R:** Changed as indicated in our answer above. See page 1, line 1.

**C9:** *-Precipitation measurements collected in valley bottoms, and assumed uniform over each sub basin. What about spatial and temporal aspects of this assumption? Please mention.*

**R:** Our subregions appear to be small enough that the climate can be considered (at least as a first assumption) as identical within. We use the average of a subregion to even out the effects of phenomena such as a very short and very localized summer thunderstorm. Yet such local effects might be responsible for some of the differences in correlations observed under flood conditions. We will consider adding this aspect to the discussion of the flood and drought years.

**R:** We have added some details on this, however we have placed it in Section 2.2 where we discuss drought indices, see page 8, line 3 ff.

**C10:**-*Page 6; line 24. "While there is some criticism of the gamma distribution (see e.g. Guttman (1999)), it is generally a widely used and recommended mended index (see e.g. Svoboda et al. (2012))."*
*Good point - there have been quite a few questions raised lately about issues of the SPI's assumed gamma distribution assumption – perhaps cite this new one and discuss – and if these findings perhaps influence yours.*
*Blain, G. C., & Meschiatti, M. C. (2015). Inadequacy of the gamma distribution to calculate the Standardized Precipitation Index.áaRevista Brasileira de Engenharia Agrícola e Ambiental,áĂa19(12), 1129-1135. http://www.scielo.br/scielo.php?pid=S1415-43662015001201129&script=sci_arttext*

**R:** Thank you for providing this reference. The authors raise some relevant issues regarding the gamma distribution and its fit (or lack thereof) for local precipitation. However, their results for precipitation in Brazil are not necessarily applicable to precipitation in Austria. Since assessing the suitability of indices is not the focus of our paper, we do not want to discuss this issue further. But since the reference provides the interested reader with a wealth of further information on this subject, we will add it to the paragraph.

**R:** Changed as indicated in our answer above. See page 8, line 21.

**C11:**-*Page 3; line 8. "For these three subregions monthly groundwater levels as well as river stages and precipitation are available at a the ehyd.gv.at website (BMLFUW, 2016)."*
*How are groundwater levels measured – steel tape, e-tapes, pressure transducers? What type of precipitation data (i.e. snow, rain, both), and how is it measurements? Please provide more background and details. Very impressive that all this historical data exists, but more background would be great to get a better picture of how it is done and perhaps other agencies can adopt.*

**R:** See also C15 by reviewer 1. We will add some more information on the Austrian ehyd.gv.at site and the underlying database.

**R:** We have added some further information and sources on the development and management of the ehyd.gv.at system to section 2.1. See page 3, line 11 ff.

**C12:** *-Page 8; line 10. "For each possible combination of standardized wells (SGI), standardized*
*precipitation (SPI) or standardized river stages (SRSI) a Pearson Correlation*
*coefficient was calculated. In order to facilitate the comparison of standardized*
*groundwater levels, river stages, and precipitation within the individual subregions, the*
*correlations between the indices have been plotted within a matrix, showing all the*
*groundwater monitoring wells, all the river stages and SPI1, 3, 6, 9 and 12 for each*
*subregion, similar to the matrices applied in Stoll et al. (2011) and Loon and Laaha*
*(2015)."*
*More details need to be discussed so the reader can get a better understanding of the*
*matrix plots – like "for example, along the x-axis is . . .. where each cell represents*
*a different . . ." etc.. Perhaps follow the descriptions by Stoll and Loon and Laaha to*
*provide further explanations.*

**R:** Please see C3 by reviewer 2 and C8 by reviewer 1.

**C13:** *-Page 8; line 16 – "According to Vekerdy and Meijerink (1998), correlations between*
*daily river stages and groundwater levels in distances similar to those relevant for this*
*paper are mostly below 30 days. Likewise, Bloomfield and Marchant (2013) as well*
*as Kumar et al. (2016) found with few exceptions the highest correlation between SGI*
*and SPI associated with a time lag of zero months. As this is particularly expected in*
*shallow alluvial aquifers, only Pearson Correlation coefficients without a time lag are*
*considered here."*
*For all wells? Did you explore if optimal time scales were greater than or less than 1*
*month? This would be a fairly easy thing to explore, especially put into context of well*
*distance from the stream, etc. Presenting correlations without time lags seems odd.*

**R:** According to the literature we cited, time lags are expected to be small to none. As this was confirmed for several samples where we calculated them, we decided to use consistently the correlation with zero time lag. We will evaluate further samples and add more information on the results of the samples.

**R:** We have calculated the correlations with lag times from 0 to 48 months for all of our data and found that our samples and our assumption where correct, meaning that we mostly have the highest correlations for a zero month lag. Thus we added a summary of this, some data and a discussion of lag times to Section 2.3. See page 9, line 25 ff.

**C14:** *-Figure 2. This is really complex to interpret. Is there any additional labeling / symbology*
*that could help? For example I count 5 boxes under the SPI 1-12, and SPSI has 3*
*boxes. Can you help the reader interpret the figure better? For example the diagonal –*
*can you tell the reader what this means? What do the "bulls eyes" correlation features*
*represent? The figure caption should be pretty self-descriptive, so this one could be*
*long – or integrated into text.*

**R:** See also C3 and C12 by reviewer 2 and C8 by reviewer 1. We suggest that we add some more information on how to read a correlation matrix in a further appendix to the text, so that the main body stays compact for those that are familiar with this method, yet those who are not can find a description of them.

**R:** We added the mentioned appendix to our manuscript. See appendix A.

**C15:** *--Page 10; line 6. "Most wells outside of the core of this region show a similar behavior,*
*resulting in an average Pearson correlation coefficient of all of these wells with each*
*other of 0.59. These wells show a low correlation with the SPI1 and moderate to*
*high correlations with the longer SPI averaging periods, as expected from the previous*
*literature (Bloomfield and Marchant, 2013; Kumar et al., 2016)."*
*Perhaps add that this is to be expected since these wells are further away?*

**R:** This is indeed to be expected, however there are also some wells in this dataset that are from a deeper aquifer. We will rephrase this section, to clarify on these two aquifers.

**R:** Changed as indicated in our answer above. We have also added markers to Figures 2 - 5, to help identifying the wells in the deeper aquifer. See page 10, line 13 ff.

**C16:**-*Page 11; line 6. The wells from the deeper aquifer also show a clear increase in correlation with an increase in the length of the SPI averaging periods, starting with an average correlation of the wells with the SPI1 of -0.04, reaching a maximum correlation of 0.38 with the SPI12, which is significantly lower than the correlations seen in the shallow wells. The average correlations of the deeper wells with the rivers range from -0.13 with the local Pöls to 0.24 with the downstream Mur.*
*Which deep well/labels should the reader be looking at exactly in Figure 2 to see this longer / smoother water level signal?*

**R:** The deep wells are represented by well AAn. We will add this information to the text and add some more thorough labelling to the figures (see also C14 above).

**R:** We have done some rewriting of the sections mentioned in the comments above and tried to generally clarify where and why we are using the deep and shallow wells from the Aichfeld subregion. Additionally, all of the deep wells are now marked in Figures 2 – 5. See page 10, line 33 ff.

**C17:**--*Page 11; line 11. "The rivers are correlated well with each other, indicating a similar flow regime in the upstream and downstream Mur, as well as in the tributary Pöls, but the correlations with the precipitation are low to moderate, ranging form an average of 0.27 with SPI1 to 0.48 with SPI6."*
*-Perhaps the low correlation is the assumption of a standardized time scale of 0 months for stream stage?*
*Can you describe the impact of snow accumulation and melt, and possible lag times in this statement?*
*See McEvoy et al. (2012) and Abatzoglou et al. (2014) on evaluating precipitation and streamflow indices for different time scales.*
*McEvoy, D. J., Huntington, J. L., Abatzoglou, J. T., & Edwards, L. M. (2012). An evaluation of multiscalar drought indices in Nevada and Eastern California. ̆aEarth Interactions, ¢a16(18), 1-18.*
*Abatzoglou, J. T., Barbero, R., Wolf, J. W., & Holden, Z. A. (2014). Tracking interannual streamflow variability with drought indices in the US Pacific Northwest.¢aJournal of Hydrometeorology,¢a15(5), 1900-1912.*

**R:** See also C11 by reviewer 1: It is generally expected that the runoff process transforms the precipitation signal, which may result in low correlation of river stage with precipitation even if lag times are considered. Nevertheless, when checking lag times for selected samples (see C13 above) we will also consider river stages. In addition, we will also discuss some aspects of snow cover and melt in our region (see C2 by reviewer 1). The stream stage is analyzed here analogously to the groundwater level and therefore "not accumulated over specific time periods due to the continuous nature of the underlying ground-water level (Bloomfield and Marchant, 2013)" (see p. 7, l. 4-5). Accumulated (or averaged) stream stages might result in higher correlation e.g. with SPI6; however, we think that this contributes little to the main questions addressed by the manuscript.

**R:** We also included river stages in our lag time analysis and found that over 95% of possible SRSI-SGI pairings have their highest correlation at a zero month time lag (see Section 2.3, page 10, line 4 ff.). We have added a thorough discussion of snow to the new Sections 3.3 and 4.3. These new additions do not impact our statement made in the line mentioned above.

**C18:**-*Page 12; line 1. "Surprisingly, some of the wells closest to the Mur on both sides of the river are not very well correlated with each other and are also not among the wells with highest correlations with the rivers."*
*Is this due to differences depth of well screen intervals / depths? Please explain.*

**R:** The wells in question are well MKr and an unnamed (in our manuscript) well directly next to well MKr but on the opposite site of the river. The cause for this difference is the construction

of a run of river power plant in this location, discussed in section 4.4. We generally try to describe in section 3 and discuss causes in section 4, but we consider adding a remark referring to section 4.4 to clarify this.

**R:** We have added a remark to well MKr earlier in the text (see page 14, line 2)and added a referrer to section 4.5 to the end of Section 3.1.2 to point the reader to the discussion of the phenomena that are solely described in this Section.

**C19:-**Page 12; Line 14. *"Well MKr is located closest to the river Mur, yet it shows no high correlation with it."*
*Why is this? What is the well depth of MKr? Can you tell the reader what your idea is about why this is?*

**R:** Wells MKr and MDp are located close to a power plant, built from 1995 to 1998. We decided to first describe our observations, and then to discuss the cause in section 4.4 but will consider referring the reader to section 4.4. See also comment above.

**R:** As indicated in the previous response the reader is now referred to section 4.5 (formerly 4.4), see page 14, lines 16 -17.

**C20:-**Page 12; line 18. *"The rivers are very highly correlated with each other, but only show some minor correlations with the 3 and 6 month SPI with average correlation coefficients of 0.38 and 0.39."*
*Perhaps because monthly correlations were not considered? See figures 9 and 10 and respective discussions in McEvoy et al. (2012) for some ideas to further describe why correlations were low, or if analyzed a different way, may increase. https://www.researchgate.net/publication/236687241_An_Evaluation_of_Multiscalar_ Drought_Indices_in_Nevada_and_Eastern_California*

**R:** The average correlation of the rivers with the 1 month SPI is even lower (0.26), as is the correlation with SPI 9 (0.35) and 12 (0.37). For the sake of brevity we decided to only mention the two highest average correlations. We suggest changing this sentence to "The rivers are very highly correlated with each other, but only show some minor correlations with SPI1 (average correlation coefficient 0.26) and SPI3 – 9 (average correlation coefficients 0.35 – 0.39)".
Regarding the mentioned paper, please refer to our answer to C17.

**R:** Changed as indicated in our answer above. See page 14, lines 19 – 20.

**C21:-**Page 16; line 2. *"Figure 4 shows the development of the three subregions when splitup into time periods of 12 years."*
*How so? Please further explain figure 4 to this point.*

**R:** The time series are split up into the mentioned 12 year periods and then the correlation coefficients making up the matrices are calculated on the basis of these 12 year periods. We will add this to the appendix about the method mentioned in our other comments.

**R:** Changed as indicated in our answer above. See page 19, lines 8 -10 and appendix A.

**C22:-**Page 17; line 29. *"At high water levels, the river feeds the groundwater, thus superpositioning its signal onto the groundwater, whereas the groundwater provides the river baseflow in low water conditions, thus giving the river a groundwater signal at low water levels."*
*Can you show this point with the standardized time series? It is not clear from the correlation matrix plots. What is the "groundwater signal at low water levels." Is there a "signal" during this condition?*

**R:** The idea here is that we have a receiving stream under dry conditions and a feeding stream under flood conditions. Partly this can be seen in figure 2, where many of the shown wells show a behavior similar to the streams. We will add some further detail on this issue.

**R:** We have reworded parts of the paragraph (see Page 23, line 31) and added a reference to section 4.2 where we discuss river-groundwater interaction in more detail, including a discussion of aquifer response times.

**C23:**-*Page 18; line 25 – suggest using the term groundwater discharge rather than groundwater runoff.*

**R:** We decided to use the term "groundwater runoff" since to stay close to the cited passage from Eltahir and Yeh, but we agree that "groundwater discharge" is a more fitting term.

**R:** Changed as indicated in our answer above (using "discharge" now).

**C24:**-*Figure 5. What time scale are the SPI values – 0 month? Please specify.*

**R:** We are working with the 1 month SPI. We will add some further information on this to the revised figures. See also C12 by reviewer 1.

**R:** Changed as indicated in our answer above. See page 25, line 12.

**C25:**-*Figure 5 and 6. Why not use 48 or 60 month time scales for SGI, SPI, and SRSI rather than using a 0 month and then smooth using a 5 year moving average of the indices? Seems a bit odd and misses the point of the use of multi-temporal time scale indices. Please explain.*

**R:** Since only the SPI is generally used for multiple time scales and we are only looking for long term trends instead of single events in these specific plots, we deemed it adequate to use a moving average on the precalculated indices.

**C26:**-*Page 22; line 4 – change hydro(geo)ology to hydrogeology. Also, it is not agreed that the "general consensus is that hydrogeology is stationarity." Milly et al., 2008 states that stationarity it is dead – but that in water management stationarity is often assumed. . . this is not the same as a "general consensus in hydrogeology seems to be the assumption of stationarity" as stated - please revise.*

**R:** We deemed this a fitting statement, considering that Milly et al. do acknowledge this assumption and the general prevalence of stationarity in textbooks and engineering decisions, since it seems to be the motivation for their letter. Also the two cited works by Koutsoyiannis are arguing for the idea of stationarity. We suggest that we tone down the statement a bit, and provide some more context.

**R:** We have reworded that part (see page 30, line 12 ff.) and added an extensive discussion of the cited Koutsoyiannis (2011) paper to this section and suggest further investigations of nonstationarity in Section 5.

**C27:**-*Page 24; line 3. "It was shown that the correlation matrix approach enables a quick visualization and comparison of different locations and time spans and that standardized indices, such as the SPI, the SGI and the SRSI (SGI applied to river levels), allow for a thorough comparison of groundwater wells, rivers and precipitation."*
*More thorough labeling and discussion on how to interpret the matrix plots when they are introduced, and during results discussion is needed for these plots to "enable a quick visualization and comparison of different locations and time spans" – please provide more details to help the reader digest these plots.*

**R:** See the other comments on this matter. We will add an appendix explaining the method in detail to those not familiar with it.

**R:** Changed as indicated in our answer above.

[revised manuscript text omitted]